# Mercury records covering the past 90 kyr from lakes Prespa and Ohrid, SE Europe

Alice R. Paine[1*], Isabel M. Fendley[1], Joost Frieling[1], Tamsin A. Mather[1], Jack H. Lacey[2], Bernd Wagner[3], Stuart A. Robinson[1], David M. Pyle[1], Alexander Francke[4], Theodore R Them II[5], Konstantinos Panagiotopoulos[3]

[1]Department of Earth Sciences, University of Oxford, Oxford, UK, OX1 3AN
[2]National Environmental Isotope Facility, British Geological Survey, Nottingham, UK
[3]Institute of Geology and Mineralogy, University of Cologne, Cologne, Germany
[4] Discipline of Archaeology,  College of Humanities, Arts and Social Sciences, Flinders University, Adelaide, 5001, Australia
[5]Department of Geology and Environmental Geosciences, College of Charleston, Charleston, SC 29424, USA
*Corresponding Author: alice.paine@earth.ox.ac.uk

## ABSTRACT

The element mercury (Hg) is a key pollutant, and much insight has been gained by studying the present-day Hg cycle. However, many important processes within this cycle operate on timescales responsive to centennial to millennial-scale environmental variability, highlighting the importance of also investigating the longer-term Hg records in sedimentary archives. To this end, we here explore the timing, magnitude, and expression of Hg signals retained in sediments over the past ~90 ka from two lakes, linked by a subterranean karst system: Lake Prespa (Greece/North Macedonia/Albania) and Lake Ohrid (North Macedonia/Albania).  Results suggest that Hg fluctuations are largely independent of variability in common host phases in each lake, and the recorded sedimentary Hg signals show distinct differences first during the late Pleistocene (Marine Isotope Stages 2 – 5). The Hg signals in Lake Prespa sediments highlights an abrupt, short-lived, peak in Hg accumulation coinciding with local deglaciation. In contrast, Lake Ohrid shows a broader interval with enhanced Hg accumulation, and, superimposed, a series of low-amplitude oscillations in Hg concentration peaking during the Last Glacial Maximum, that may result from elevated clastic inputs. Divergent Hg signals are also recorded during the early and middle Holocene (Marine Isotope Stage 1). Here, Lake Prespa sediments show a series of large Hg peaks; while Lake Ohrid sediments show a progression to lower Hg values. Around 3 ka, anthropogenic influences overwhelm local fluxes in both lakes. The lack of coherence in Hg accumulation between the two lakes suggests that, in the absence of an exceptional perturbation, local differences in sediment composition, lake structure, Hg sources, and water balance all influence the local Hg cycle, and determine the extent to which Hg signals reflect local or global-scale environmental changes.

## 1. Introduction

Mercury (Hg) is a volatile metal released into the environment from both natural and anthropogenic sources, and actively cycled between surface reservoirs (e.g., atmosphere, ocean, lakes). Emissions

of Hg by geological processes are unevenly distributed across the Earth's surface, and are generally concentrated where tectonic, volcanic, and geothermal activities are most intense (Rytuba, 2003; Edwards et al., 2021; Schlüter, 2000). Geological processes have been major drivers of variability in the global Hg cycle throughout Earth's history (Selin, 2009), leading to the use of sedimentary Hg to reconstruct periods of intense volcanism (e.g., large igneous provinces (LIPs)) in Earth's geological past (e.g., Grasby et al., 2019; Percival et al., 2018). In recent times, Hg release associated with industrialisation, the extraction and combustion of fossil fuels, and natural resources (metals) has overwhelmed the natural background flux (Outridge et al., 2018; Streets et al., 2019; United Nations Environment Programme, 2018).

Existing in the atmosphere primarily in the form of gaseous elemental mercury, Hg has an atmospheric lifetime of up to 2 years, facilitating its deposition far from the original source (Lyman et al., 2020). Once removed from the atmosphere, Hg may enter vegetation and soils where it is cycled between reservoirs by a complex series of processes, many of which occur on timescales that exceed present-day monitoring (**Fig. 1**) (Branfireun et al., 2020; Selin, 2009). Evasion back to the atmosphere, consumption by living organisms, or sequestration within aquatic sediments all represent ways in which Hg may 'leave' the terrestrial environment, and aquatic sediments are known to be particulary effective sinks within the global Hg cycle (Bishop et al., 2020; Selin, 2009). Here, microbial processes lead to the formation of methylmercury (MeHg), which is the most bio-accumulative Hg species and can cause severe neurological and physiological damage to complex organisms if ingested (Driscoll et al., 2013; Wang et al., 2019).

The ecological and societal risks of environmental Hg contamination underscore the importance of quantifying how natural and anthropogenic processes may influence Hg sequestration within aquatic systems, and the timescales upon which they are effective. Time-resolved sediment records sourced from marine and lacustrine basins are highly suitable for assessing these roles further back in time, as the Hg deposited may originate from one of several potential sources in the atmospheric (e.g., precipitation, dust), terrestrial (e.g., soils, detrital matter), aquatic, and/or lithospheric domain (**Fig. 1**). Thus, they can provide time-resolved records of Hg deposition, cycling, burial, and accumulation relative to changing environmental conditions on a local, regional, or even global-scale (Cooke et al., 2020; Zaferani and Biester, 2021), and so can offer new insights into the cycling of Hg in the terrestrial realm.

Analysis of pre-industrial marine and lacustrine sediment records suggest that Hg concentration broadly reflects variability in climate (Li et al., 2020). On orbital ($>10^3$-year) timescales, oceanic Hg signals manifest as low-amplitude fluctuations corresponding to global-scale climate shifts from warm (interglacial) to colder (glacial) conditions; for example due to changes in atmospheric composition (e.g., mineral dust loading) and circulation, biogeochemical cycling (Figueiredo et al., 2022), and/or ocean circulation (Figueiredo et al., 2020; Gelety et al., 2007; Jitaru et al., 2009; Kita et al., 2016). On centennial to millennial ($10^2$-$10^3$-years) timescales, lacustrine Hg signals correspond more closely to transient changes in hydrology, landscape dynamics, and ice/permafrost extent on local/regional scales (Chede et al., 2022; Cordeiro et al., 2011; de Lacerda et al., 2017; Fadina et al., 2019; Li et al.,

2023; Pérez-Rodríguez et al., 2018, 2015) (**Fig. 1**). Importantly, climate-associated Hg signals
retained in lacustrine records integrate a range of processes and some records show higher
sedimentary Hg concentrations during cold, arid conditions (e.g., Li et al., 2020), while other records
tend to have higher Hg concentrations with warm and wet climates. For example, increases in
catchment-sourced detrital input have been proposed as the primary cause of Hg enrichment in
temperate lakes (Pan et al., 2020; Schütze et al., 2018), and near-shore marine records (Fadina et
al., 2019). Conversely, lakes located in glaciated regions may show dilution of Hg by the same inputs
(Schneider et al., 2020). Local, site-specific factors are therefore likely to influence sedimentary Hg
records. Yet, the combined effects of global and local processes complicate study of how changes in
the terrestrial Hg cycle may translate to measurable sedimentary signals and signals that are
comparable between different regional or global archives.

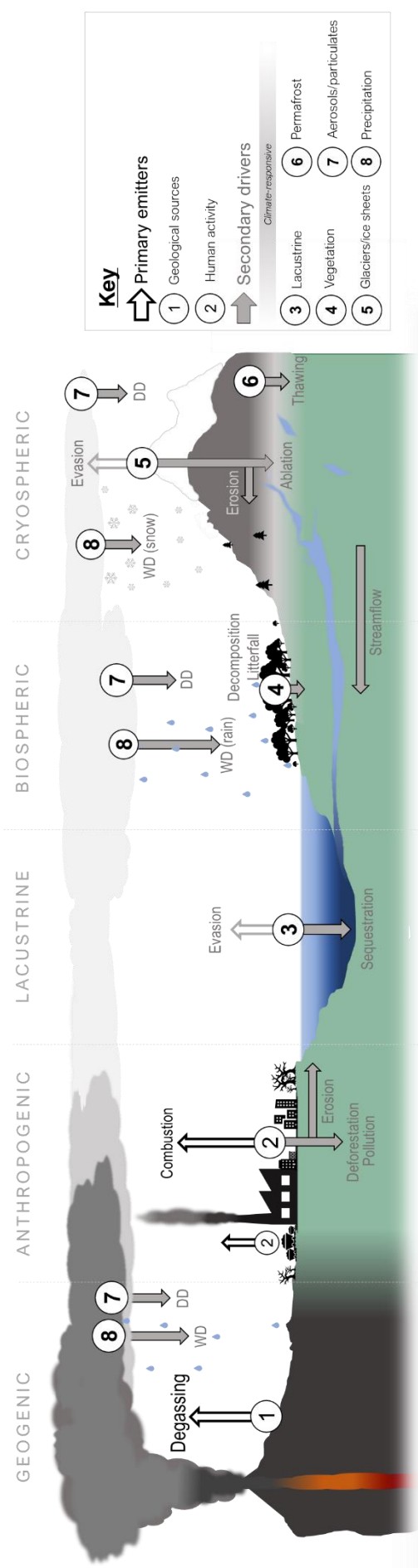

**Figure 1**: A summary diagram depicting the key anthropogenic, geogenic, biospheric, cryospheric, and lacustrine processes, which could generate and modify a sedimentary Hg signal over $10^1$-$10^5$-year timescales. Processes are abbreviated as: WD – wet deposition, DD – dry deposition. Non-filled arrows depict processes acting to increase the atmospheric Hg burden, and colour filled arrows depict processes acting to influence the quantity of Hg stored in terrestrial reservoirs. This figure is schematic (not drawn to scale), and constructed on the basis of reviews by Bishop et al. (2020), Obrist et al. (2018), Selin et al. (2009).

Sedimentary Hg presence (or absence) at discrete intervals can be quantified using the total Hg
concentration ($Hg_T$) (Bishop et al., 2020; Kohler et al., 2022; Nasr et al., 2011). However, internal
changes in bioproductivity, organic matter type and/or flux, sedimentation rate, pH, and redox
conditions could all produce a distinct, local, transient, sedimentary Hg enrichment without a
meaningful change in the total amount of Hg present and/or mobile in the broader aquatic system. In
light of these complexities, it has become common practice to examine total Hg concentration ($Hg_T$)
alongside Hg concentration divided by (normalised to) the concentration of various chemical species.
Normalisation is often applied when it can be shown that the abundance of a carrier (or "host") phase
directly impacts Hg content. Normalisation (e.g. Hg/total organic carbon (TOC), Hg/total sulphur (TS))
may, in those cases, then reveal broader changes in environmental Hg availability (Grasby et al.,
2019; Percival et al., 2015; Shen et al., 2020; Them et al., 2019). Such an approach is particularly
beneficial for studies typically spanning $>10^2$-year timescales, where the goal is to isolate the effects
of catchment-scale depositional and/or transport processes on Hg signals recorded in the sediment
through time.
Organic matter (hereafter represented by total organic carbon (TOC)) is generally considered the
dominant carrier phase of sedimentary Hg (Chakraborty et al., 2015; Ravichandran, 2004). For
records in which TOC and Hg co-vary linearly, Hg is generally normalized to TOC (Chede et al., 2022;
Figueiredo et al., 2022, 2020; Kita et al., 2016; Outridge et al., 2019). Some systems do not exhibit a
relation to TOC and Hg may instead be adsorbed onto (fine-grained) detrital minerals and detected by
a correlation between Hg and mineral-dominating elements such as aluminium (Al), titanium (Ti),
zirconium (Zr), rubidium (Rb), or potassium (K) (Sanei et al., 2012; Sial et al., 2013; Them et al.,
2019). In few cases, sulphide minerals may act as important Hg hosts (Benoit et al., 1999; Han et al.,
2008), however this is less common in freshwater lacustrine systems where sulphate-reduction is
often limited and only a small fraction of non-organic sulfur is buried (Ding et al., 2016; Holmer and
Storkholm, 2001; Tisserand et al., 2022; Watanabe et al., 2004).
Mercury's relationship with other sedimentary components is often complex. For example, $Hg_T$ may
also be suppressed through dilution by Hg-poor detrital or biogenic (carbonate, silica) material, and
Hg in many sediments is not exclusively or clearly modulated by balances between host-phase
abundance and dilution. Notably, this can also occur when the host-phases are always present in
sufficient quantities to sequester available Hg. In such cases, and where (single) host-phase
abundance or dilution cannot be easily accounted for, Hg accumulation rate ($Hg_{AR}$) may provide the
most optimal assessment of Hg availability through time as long as a robust age model is available for
the archive.
Sedimentary TOC, total sulphur (TS), and detrital and biogenic mineral concentrations change in
space and time, underscoring the need to assess how Hg covaries in relation to different host phases
and other sedimentary materials. Hydrology, sedimentation regime, and geochemistry may each
influence mercury host-phase availability and burial in a lacustrine system, and are likely to change
through time, highlighting the importance of investigating the longer-term records of Hg burial and
accumulation in sedimentary archives.
This study explores the timing, magnitude, and expression of Hg signals retained in the sediment
records of Lake Prespa (Greece/Albania/North Macedonia) and Lake Ohrid (North
Macedonia/Albania) over the past ~90 ka. The two lakes are located only ~10 km apart (**Fig. 2**), are
hydrologically connected by karst aquifers with ~50% of water inflow to Lake Ohrid originating from
Lake Prespa (Matzinger et al., 2006), and their sediments encode records of environmental change in
southeast Europe over the last ~90 ka (Damaschke et al., 2013; Francke et al., 2016; Leng et al.,
2010; Panagiotopoulos et al., 2014; Sadori et al., 2016; Wagner et al., 2010).  Comparison of their
sedimentary records provides a rare opportunity to explore three important questions. First, we test
how the local sedimentary environment (e.g., host phase availability and sources) influences Hg
burial. Second, we investigate whether Hg signals reflect changes in catchment hydrology, structure,
and/or varying degrees of interaction between the two lake systems. Finally, we explore whether
regional-scale climate variability could have measurably affected the Hg signals retained in the
sediments.

# 141  2. Site Description

## 142  2.1.   Regional Climate

The Mediterranean Sea and the European continent are both major influences on present-day climate
of the region surrounding lakes Prespa and Ohrid. Summer months (July to August) are hot and dry
(average monthly air temperature +26 °C) while winter months (November to January) are cold,
cloudy and wet, with an average monthly air temperature of −1 °C (Matzinger et al., 2006). Annual
precipitation in the region averages ~750 mm yr$^{-1}$, with winter precipitation falling predominantly as
snow at high elevations (Hollis and Stevenson, 1997). Present-day vegetation in the Prespa/Ohrid
region comprises a mixture of Balkan endemic, central European, and Mediterranean species
(Donders et al., 2021; Panagiotopoulos et al., 2014, 2020; Sadori et al., 2016).
Major shifts in sedimentation and catchment structure of lakes Prespa and Ohrid generally
correspond to the large-scale climate oscillations captured by proxy records across southern Europe
throughout the last glacial-interglacial cycle (~100-kyr) (e.g., Rasmussen et al., 2014; Sanchez Goñi
and Harrison, 2010; Tzedakis et al., 2006). Generally higher local temperatures and moisture
availability are observed during the last interglacial (pre-74 ka), following which conditions became
distinctly colder and/or drier. This resulted in the rapid recession of forest ecosystems, intense erosion
of local soils and catchments, and elevated aeolian activity (e.g., Panagiotopoulos et al., 2014; Sadori
et al., 2016; Francke et al., 2016). Although slightly warmer conditions were restored between ~57
and 29 ka, both moisture availability and temperature dropped again during the Last Glacial Maximum
(LGM; ~29 – 12 ka) – favouring the growth and development of glaciers and (peri)glacial features
(e.g., moraines) in the Prespa/Ohrid catchment (Ribolini et al., 2018; Gromig et al., 2018; Ruszkiczay-
Rüdiger et al., 2020), but also across the Balkan peninsula (Allard et al., 2021; Hughes and
Woodward, 2017; Leontaritis et al., 2020). Lake Prespa's sediments host evidence for millennial scale
climate varaiblity during the Last Glacial, which were tentatively correlated to Heinrich Events in the

North Atlantic (Wagner et al., 2010). At ~12 ka, the Pleistocene to Holocene transition saw the rapid propagation of warmer, wetter conditions across the region (known as Termination I) with only brief excursions from this warming trend, such as episodes of transient drying and/or cooling at 8.2 ka and 4.2 ka (Bini et al., 2019; Aufgebauer et al., 2012a). Anthropogenic influence on the Balkan landscape becomes increasingly clear from ~2.5 ka onwards, mainly in the form of increased erosion regimes, forest clearance, agricultural land modification, and evidence for metallurgic practices (Panagiotopoulos et al., 2013; Cvetkoska et al., 2014; Radivojević and Roberts, 2021).

## 2.2. Lake Prespa

The Prespa lake system (40°54' N, 21°02' E) is composed of two lakes separated by an isthmus and located on the tripoint of North Macedonia, Albania and Greece, at an altitude of 844 metres (m) above sea level. The ~1300 $km^2$ catchment of the Prespa lakes encompasses the Pelister Mountains to the east and the Galiçica Mountains to the southwest and west (**Fig. 2**). Here we focus on Megali Prespa (hereafter referred to as Lake Prespa), the larger of the two lakes, which has a surface area of 254 $km^2$, a maximum water depth of 48 m, and a mean water depth of 14 m. The total inflow into Lake Prespa averages ~16.9 $m^3$ $s^{-1}$ (Matzinger et al., 2006). Water input is sourced from surface runoff (56%), direct precipitation (35%), and inflow from the smaller of the two lakes (Mikri Prespa; 9%) (Matzinger et al., 2006). Lake Prespa has no surface outflow. The residence time of the lake's waters is ~11 years (Matzinger et al., 2006) and water is predominantly lost through evaporation (52%), underground karst channels into Lake Ohrid located 10 km to the west (46%), and irrigation (2%). The lake is currently mesotrophic with an average total phosphorus (TP) concentration of 31 mg $m^{-3}$ in the water column, basal anoxia in summer months, and generally clear waters; all signalling moderate biological productivity (Hollis and Stevenson, 1997). However, the lake likely held a more oligotrophic (low) nutrient status during the colder late Pleistocene, where biological produciy reduced substantially (Matzinger et al., 2006; Wagner et al., 2010).

## 2.3. Lake Ohrid

Lake Ohrid (41°02′ N, 20°43′ E) lies 693 m above sea level. Separated from Lake Prespa by the Galiçica Mountains, the lake straddles the boundary between North Macedonia and Albania (**Fig. 2**). The lake is ~30 km long and 15 km wide, with a maximum water depth of 293 m, water volume of 55.4 $km^3$, and hydraulic residence time of ~70 years. Water input is sourced from direct precipitation (23%), river inflow (24%), and karst springs (53%) fed by precipitation and water from Lake Prespa (Matzinger et al., 2006; Lacey and Jones, 2018), and this hydrological link increases the Ohrid catchment by ~1300 $km^2$ to ~2610 $km^2$. Evaporation (40%) and outflow via the river Crn Drim (60%) are the dominant pathways for water loss from Lake Ohrid, and complete mixing of the lake occurs only every few years (Matzinger et al., 2006). The present-day lake shows low levels of biological

201  productivity (oligotrophic) with an average dissolved phosphorus content of 4.5 mg m$^{-3}$, and regular

202  mixing maintains moderately oxygenated bottom waters (Matzinger et al., 2006; Wagner et al., 2010).

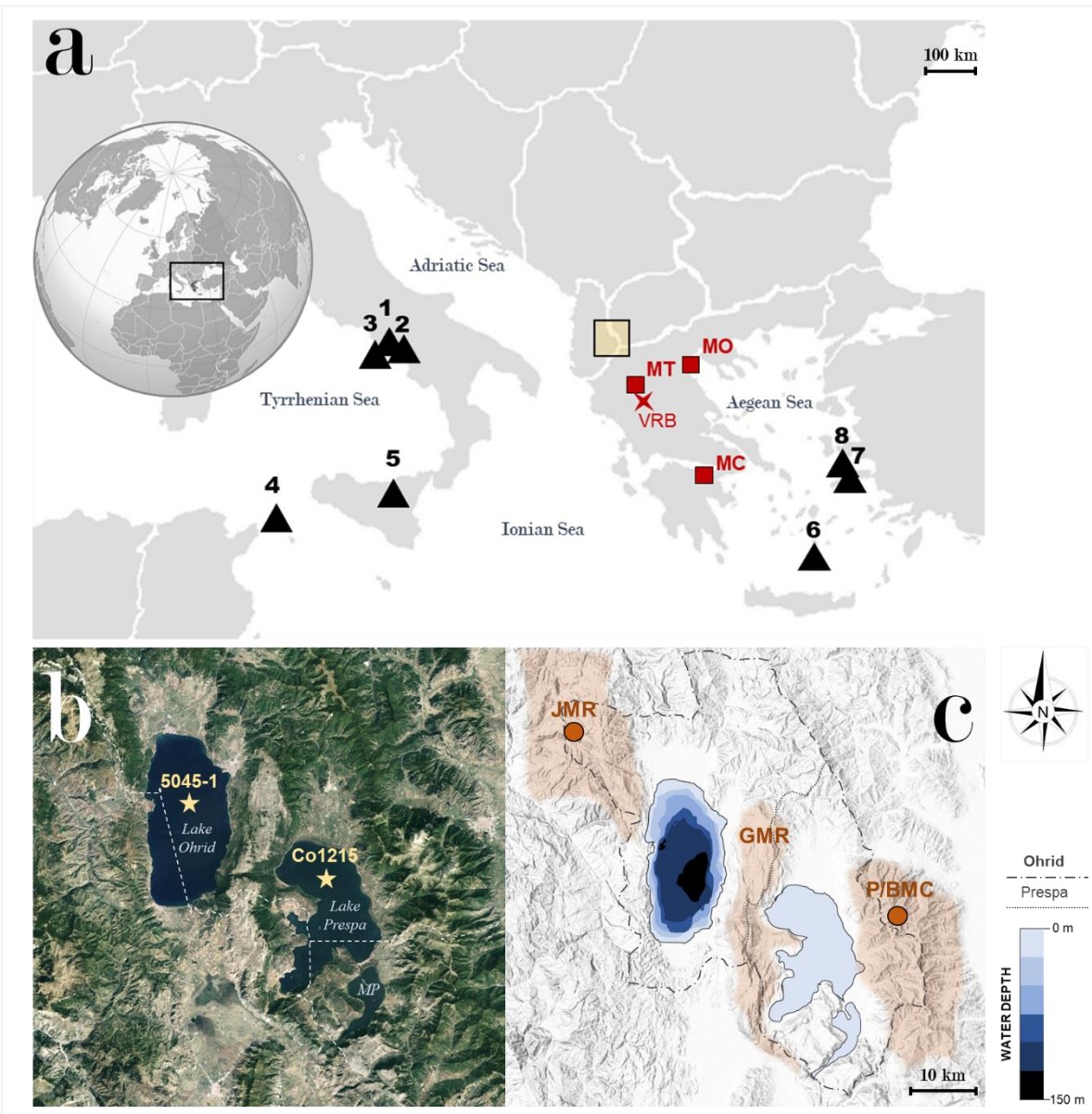

**Figure 2**: (a) Map showing the location of lakes Prespa and Ohrid within Southern Europe (yellow shaded box). Volcanoes from which tephra has been identifed in Co1215 (Prespa) and/or 5045-1 (Ohrid) are coloured as black triangles, and numbered as: 1 – Vesuvius, 2 – Campi Flegrei, 3 – Ischia, 4 - Pantelleria, 5 – Etna. Volcanoes of the South Aegean Volcanic Arc with known explosive eruptions (>magnitude 4.0) between 90 and 0 ka are also numbered: 6 – Santorini, 7 – Nisyros, 8 – Yali. Sites referred to in this study are also labelled as follows: (red squares) MT – Mount Tymphi, MO – Mount Olympus, MC – Mount Chelmos; (red star) VRB – Voidomaitis river basin.  (b) Aerial photo showing the coring locations of Co1215 and 5045-1, and illustrating the vegetation distributions of the area surrounding lakes Prespa and Ohrid. Mikri Prespa is labelled as 'MP' Base image sourced from GoogleEarth v 9.177.0.1$^{TM}$. (c) Hillshade map of the Prespa/Ohrid region and bathymetric data of lakes Prespa and Ohrid (Jovanovska et al., 2016; Wagner et al., 2022). Grey dashed lines denote watershed boundaries for lakes Prespa and Ohrid , respectively adapted from Panagiotopoulos et al. (2019). Basemap sourced from ArcGIS v 10.0$^{TM}$ (spatial reference 102100 (3857)). Orange shading denotes mountain ranges are labelled as: P/BMC – Pelister/Baba mountain chain (circle marking the location of Mount Pelister: 2601 m a.s.l), GMR – Galičica mountain range, and JMR – Jablanica mountain range (circle marking the location of Jablanica Mountain - 2257 m a.s.l). All mountain ranges contain evidence for the presence of glaciers and/or (peri)glacial features of late Pleistocene age (Hughes et al., 2022, 2023)

# 3. Methods

## 3.1. Lake Prespa (Co1215)

Composite core Co1215 was recovered in autumn 2009 and summer 2011 from the central-northern section of Lake Prespa (40°57'50" N, 20°58'41" E, **Fig. 2**). Sediment recovery was performed using a floating platform, with a gravity corer for surface sediments and a 3-m-long percussion piston corer (UWITEC Co. Austria) for deeper sediments. Overlapping 3-m-long sediment cores were cut into segments of up to 1 m in length for transport and storage. After splicing and correlation of core segments according to geocemical and optical infromation, the resulting 17.7 m composite core was continuously sampled at 2-cm-resolution, yielding a total of 849 samples. It is comprised of three major lithofacies, which differ in colour, sediment structure, grain size, organic-matter and carbonate content, and geochemistry. There are no lithological indications of any hiatuses or instances of non-contiguous sedimentation in core Co1215. A detailed lithostratigraphic characterisation of the entire succession (90–0 ka) is presented in Damaschke et al. (2013), along with details of the six visible tephra layers and five cryptotephra layers identified in Co1215 (**Table S3**).

Published data for Lake Prespa (Co1215) includes: total carbon (TC), total inorganic carbon (TIC), and total sulphur (TS) analyses (Aufgebauer et al., 2012; Damaschke et al., 2013). These data were measured at ~2 cm resolution with a DIMATOC 200 (DIMATEC Co., Germany), and TS using a Vario Micro Cube combustion CNS elemental analyser (VARIO Co.) at the University of Cologne. TOC was calculated as the difference between TC and TIC by Aufgebauer et al. (2012) for the upper ~3.2 m, and by Damaschke el a. (2013) for the full ~17 m succession. The inorganic chemistry of the sediments was determined by X-ray fluorescence (XRF) data, generated using an ITRAX core scanner (COX Ltd., Sweden) equipped with a Mo-tube set to 30 kV and 30 mA, and a Si-drift chamber detector (Wagner et al., 2012). Core Co1215 was scanned with a resolution of 2 mm and a scanning time of 10 seconds per measurement. Elemental intensities were obtained for potassium (K), titanium (Ti), manganese (Mn), strontium (Sr), iron (Fe), calcium (Ca), and rubidium (Rb) (Wagner et al., 2012).

### 3.1.1. Chronology

A chronology for Co1215 was previously produced by linear interpolation using volcanic ash layers, coupled with $^{14}$C and electron spin resonance (ESR) dates obtained for bulk organic, fish, and aquatic plant remains (Aufgebauer et al., 2012). Here, we update this chronology with a Bayesian age-depth model that re-calculates previously obtained $^{14}$C-dates (**Table S4**) with the latest (Intcal2020) radiocarbon calibration (**Fig. 3**) (Reimer et al., 2020). We used rBacon v 2.5.7 (Blaauw and Christen, 2011), and the new age model includes updated $^{40}$Ar/$^{39}$Ar dates of two eruptions geochemically correlated to specific tephra layers within the Prespa core (Damaschke et al., 2013); the Y-5 (39.85 ± 0.14 ka, 2σ (Giaccio et al., 2017)) and Y-6 (45.50 ± 1 ka , 2σ (Zanchetta et al., 2018; Scaillet et al., 2013)) tephra units. Every tephra layer is assumed to have been deposited instantaneously. The final

model used herein presents the median of all the iterations (generally indistinguishable from the
mean), and when referring to ages of specific depths within the core we include the 95% confidence
intervals. The upper 2 m (Holocene) section of core Co1215 is chronologically well constrained by 10
[14]C dates and two tephra layers, with modelled age uncertainties in this section ranging from ~5 to
580 years. Uncertainty increases with depth due to the lack of independent chronological anchors
available. For example, three ESR dates for a shell fragment layer (~14.6 m depth) give an average
age of 73.6 ± 7.7 ka, and form the only tie point currently available below 8.5 m. All twenty-seven tie-
points and accompanying chronological details are presented in **Text SI3** and **Table S3**. Our revised
model shows broad agreement with the interpolation-based chronology presented by Damaschke et
al. (2013), and suggests that core Co1215 provides a continuous record of sedimentation over the
past ~90-kyr (**Fig. S1**), with each 2 cm sample equating to ~100 years (on average).

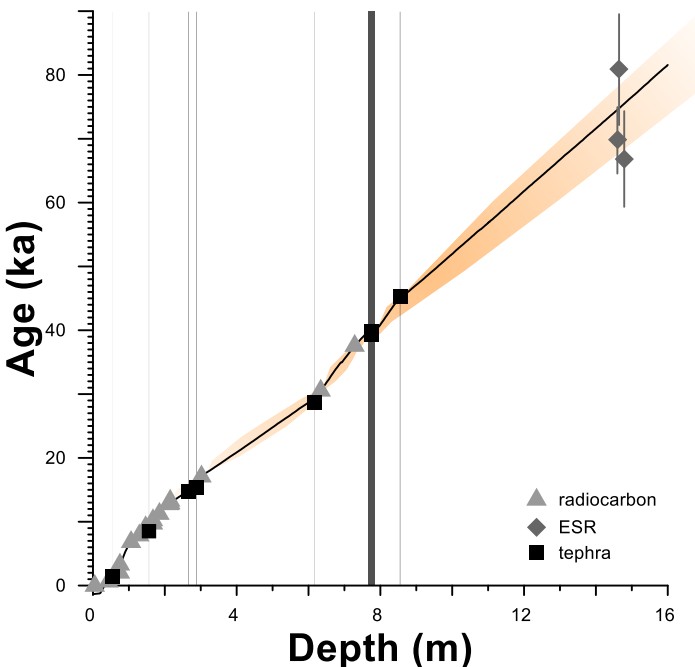

**Figure 3**: A Bayesian age-depth model for core Co1215 from Lake Prespa. Calibrated ages for the twenty-seven tie points used in model generation are displayed by type: radiocarbon-dated bulk organic, fish, or aquatic plant remains (light grey triangles), volcanic tephra layers (black squares) and electron-spin resonance (ESR)-derived dates for a shell layer (*Dreissena*) located at 14.63–14.58 m depth (dark grey diamonds). Uncertainties for ESR dates at 1σ are presented as dark grey vertical lines. Black line marks the median core age predicted by the model, which is generally indistinguishable from the predicted mean. Minimum and maximum model ages at 95% (2σ) confidence are marked with orange shading. Grey bars mark the stratigraphic placement of tephra layers used as tie-points, and widths of these bars are proportional to the thickness of the tephra layers within the core, respectively. Uncertainties for radiocarbon and tephra dates are within the displayed point sizes, and presented in **Table S4**.


## 3.2. Lake Ohrid (core 5045-1)

The 5045-1 coring site ("DEEP") is located in the central part of Lake Ohrid (41°02'57" N, 20°42'54" E)
(**Fig. 2**).  The uppermost 1.5 m of sediments at DEEP were recovered in 2011 using a UWITEC
gravity and piston corer. Sediments below 1.5 m depth were recovered from six closely-spaced drill
holes at the site in 2013 (5045-1A to 5045-1F), with a total composite field recovery amounting to >
95% (545 m); accounting for overlap between cores (Wagner et al., 2014b). Sediment cores were
spliced to a composite record using optical and geochemical information. For sedimentological and
geochemical analyses, 2 cm thick slices (40.7 cm$^3$) were removed from the core at a resolution of 16
cm (~480-yr) at the University of Cologne. For this study, we analysed 217 samples from between 0
and 36.27 m composite depth. We cannot entirely rule out that changes in sedimentation occurred
between samples, however, recent seismic (Lindhorst et al., 2015), borehole logging (Ulfers et al.,
2022) and sedimentological studies (Wagner et al., 2022, 2019) suggest that sedimentation at the
DEEP site has been near-continuous since ~1.3 Ma, with no clear evidence for any major (>1-kyr)
hiatuses. A detailed lithostratigraphic characterisation of the 5045-1 core succession is presented by
Francke et al. (2016). Details of the six microscopic and two visible tephra layers identified in the ~36
m section analysed in this study are presented by Leicher et al. (2021), and listed in **Table S5**.
The Hg data obtained from core 5045-1 (Lake Ohrid) are presented herein alongside two previously
existing datasets. The first dataset comprises TC and TIC measured using a DIMATOC 200 (TOC
calculated as the difference between TC and TIC), and TS using a Vario Micro Cube combustion CNS
elemental analyser at the University of Cologne - both by Francke et al. (2016). The second dataset
comprises XRF data obtained using an ITRAX XRF core scanner at the University of Cologne at 2.56
m increments, carried out on 2 cm thick samples, and processed using QSpec 6.5 software (Cox
Analytical) by Francke et al. (2016). Elemental intensities were obtained for K, Ti, Fe, Zr, and Ca. To
validate the quality of the XRF scanning data, conventional wavelength dispersive XRF (WDXRF,
Philips PW 2400, Panalytical Cor.) was conducted on the 2-cm-thick samples at 2.56-m resolution.
ITRAX data for each WDXRF sample was averaged to ensure comparability with the conventional
XRF data, and r$^2$ values were to compare ITRAX and WDXRX datasets (Francke et al. (2016).

### 3.2.1. Chronology

This study uses the age-depth model generated by Francke et al. (2016), and extended by Wagner et
al. (2019) for the upper ~248 m and ~447 m of core 5045-1, respectively. Both combined
tephrochronological data with orbital parameters using a Bayesian age modelling approach (Bacon
2.2). Tephra layers were used as first-order constraints. From the eleven total $^{39}$Ar/$^{40}$Ar dated tephra
layers employed in Wagner et al. (2019), seven are found in the upper ~36 m section analysed in this
study. The age of the eighth tie-point (OH-DP-0009) is defined following geochemical correlation of
this tephra layer to the AD472/512 eruption of Somma-Vesuvius, Italy (Francke et al., 2019; Leicher et
al., 2021). This chronological information was coupled with climate-sensitive proxy data (TOC and
TIC) to define cross-correlation/inflection points with orbital parameters, which were included in the
age–depth model as second-order constraints (**Table S6**). Four of these points correspond to the ~36
m interval analysed in this study (Wagner et al., 2019). The 95% confidence intervals of ages for
specific depths produced by the model average at ±5.5 kyr, with a maximum of ±10.6 kyr. The
resulting chronology suggests that the 0.97-36.27 m core section analysed here covers the time
interval 1.6 – 89.6 ka, with each sample possessing a resolution of ~400 years (Francke et al., 2016;
Wagner et al., 2019). Full description of the 5045-1 chronology and associated methods are
presented in **Supplementary Text SI4.**

## 299    3.3.    Mercury measurements

Total Hg concentrations ($Hg_T$) in the bulk sediments of cores 5045-1 (Ohrid) and Co1215 (Prespa)
were measured using an RA-915 Plus Portable Mercury Analyzer with PYRO-915 Pyrolyzer, Lumex
(Bin et al., 2001) at the University of Oxford. Samples were analysed for $Hg_T$ at a resolution of ~2 cm
for Co1215 (Lake Prespa), and ~16 cm for 5045-1 (Lake Ohrid) (see sections **3.1** and **3.2**).
Approximately 2 cm$^3$ of sediment was homogenized to fine powder for TOC (Wagner et al., 2019;
Francke et al., 2016; Aufgebauer et al., 2012a; Damaschke et al., 2013) and Hg analyses (this study).
For Hg analysis, powdered samples were weighed into glass measuring boats, with masses ranging
between 35–96 mg for Co1215, and between 27–78 mg for 5045-1. For samples particularly rich in
inorganic fractions (e.g., samples coinciding with tephra layers), masses needed to be greater in order
to yield a sufficiently high peak area (Lumex output) for calculation of sediment mercury
concentrations. Samples were then placed into the pyrolyzer (Mode 1) and heated to ~700$^o$C,
volatilizing any Hg in the sample. Spectral analysis of the gases produced yields the total Hg content
of the sample. Six measures of standard material (paint-contaminated soil – NIST Standard
Reference Material ® 2587) with an expected Hg concentration of 290 ± 9 ng g$^{-1}$ (95% confidence)
were run to calibrate the instrument prior to sample analysis, and then one standard between every
10 lacustrine samples (calibration results in **Supplementary Information**). Long-term observations of
standard measurements with total Hg yield similar to the sediment samples analysed here indicate
reproducibility is ±6 % or better for Hg concentrations >10 ng g$^{-1}$ (Frieling et al., 2023), and with Hg
recovery close to 100% as expected from pyrolysis-based instrumentation (Bin et al., 2001).  Details
of standard runs for each core are included as a supplementary file.

### 321    3.3.1.    Mercury accumulation

Rates of Hg accumulation in both cores were calculated by:
$$Hg_{AR} = Hg_T \, (DBD \times SR) \qquad \textit{(eqn. 1)}$$
where $Hg_{AR}$ is the total Hg mass accumulation rate (mg m$^{-2}$ kyr$^{-1}$), $Hg_T$ is the total mercury
concentration (expressed in mg g$^{-1}$), DBD is the dry bulk density (g m$^{-3}$), and SR is the sedimentation
rate (SR) in m kyr$^{-1}$. Values for $Hg_{AR}$ are also calculated with respect to the median age estimate for
each sample, meaning that uncertainties increase with depth.
Sedimentation rates for both Prespa and Ohrid were calculated by combining stratigraphic and
lithological observations with the age-depth relationship ascertained for each core, respectively. For
Lake Prespa, we calculate the sedimentation rate using the updated age-depth model presented in
**section 3.1.2**. Dry bulk density values were calculated on the basis of sedimentological data available
for each core. For the Lake Ohrid dataset, DBD values were already available following the analyses
of Francke et al. (2016). To acquire these values for Lake Prespa, we employed the formula:
$$DBD = M_{solid} / V_{total} \qquad \text{(eqn. 2)}$$
where $M_{solid}$ is the mass of dry solid material (g) measured in each sample, and $V_{total}$ is the volume of
each respective sample (2 cm³). Values for $M_{solid}$ were calculated based on recorded weight loss
between wet and dry samples taken for CNS analyses by Aufgebauer et al. (2012), assuming an
average wet density of 1 g cm$^{-3}$ for wet sediments, and 2.6 g cm$^{-3}$ (grain density) for dry sediments.
For Lake Ohrid, we utilise the sedimentation rate values calculated by Wagner et al. (2019), and dry
bulk density measurements measured by Francke et al. (2016) (see these publications for full
methods).

## 3.4.  Mercury normalization

The availability of specific host phases is often assumed to exert control on the sedimentary burial of
Hg. Here, we test if the Hg deposited into the sediments of lakes Prespa and Ohrid may be impacted
by abundance of a suite of phases. To do this, we assess both Hg$_T$ records relative to quantitative
estimates of TOC and TS (assuming sulphides contribute to TS): both considered potential host
phases of Hg in sedimentary successions (Chakraborty et al., 2015; Garcia-Ordiales et al., 2018;
Ravichandran, 2004; Shen et al., 2020).
Detrital minerals constitute another potential host phase of Hg in sedimentary records. Elements such
as Al, Ti, K, Zr, and Rb are commonly used as proxies for this purpose (Kongchum et al., 2011;
Percival et al., 2018b; Shen et al., 2020). We observe a close correlation between K and Ti in Lake
Prespa, and quartz in Lake Ohrid (**Fig. S2**): all proxies for fine-grained material inputs to a lake basin
(Grygar et al., 2019; Warrier et al., 2016). To facilitate direct comparison of the two cores, we assess
the relative abundances of (fine-grained) detrital material using XRF-based K counts.  To account for
differences in resolution between Hg and XRF data, K measurements were averaged to the thickness
of each discrete Hg sample, and K values corresponding to the Hg sample depths extracted.
In line with previous studies (Shen et al., 2020), we assume that the strongest positive-sloped linear
correlation with Hg among the analysed elements TS, TOC, and K signals the most likely dominant
influence on Hg loading in each core, which may then be interpreted as the 'host-phase'. However, it
is conceivable that different host phases may dominate in different sections of the individual cores or
that no single host-phase clearly dominates, and so the same approach is also applied restricted to
the data within each individual marine isotope stage (MIS) (**Table 1**).


# 4. Results & Discussion

Sediment cores extracted from Lake Prespa (Co1215) and Lake Ohrid (5045-1) provide a detailed, time-resolved record of Hg cycling between ~90 and 0 ka. Results are presented with direct reference to key stratigraphic intervals: the Holocene (12–0 ka; MIS 1), and the late Pleistocene (120 –12 ka; MIS 2–5). Widespread proxy-based evidence for warmer temperatures, forest expansion, and increased precipitation representative of interglacial climatic conditions marks the start of the Holocene epoch (~12 ka) in SE Europe (Kern et al., 2022; Panagiotopoulos et al., 2014; Sadori et al., 2016; Tzedakis et al., 2006). For simplicity, we hereafter equate "MIS 1" to the Holocene, allowing a clearer distinction between glacial (late Pleistocene) and interglacial (Holocene) climate conditions. We use these time-slices, that also represent broad climate and environmental 'modes', as a framework upon which the Hg composition of both cores can be directly compared relative to local changes in sediment lithology and geochemistry (**Table 1**), and a foundation upon which local and regional-scale environmental changes can be assessed relative to global shifts in glaciation, climate, sea level, and ocean circulation. We first consider the extent to which soft sediment processes (**section 4.1**) and lithological features (**section 4.2.**) may have influenced the Hg variability observed in **Figures 5** and **6**, before adopting a catchment-scale perspective in **section 4.3** to explore the role of diverse environmental processes in Hg cycling through these two systems.

**Table 1**: A comparison of the features of cores Co1215 (Lake Prespa) and 5045-1 (Lake Ohrid) relative to the late Pleistocene (LP; 120 – 12 ka), the Holocene (H; 12 – 0 ka), and the marine isotope stage (MIS) stratigraphic framework defined in Lisiecki & Raymo (2005)*. $Hg_T$ is given in ng g$^{-1}$, and $Hg_{AR}$ is given in mg m$^{-2}$ kyr$^{-1}$.

| | | | Depth (m) | Mean | | Sedimentology** | |
| --- | --- | --- | --- | --- | --- | --- | --- |
| | | | | $Hg_T$ | $Hg_{AR}$ | Lithology | Key Features |
| Lake Prespa | Holocene | MIS 1 | 2.4–0 | 64.6 | 11.9 | Silt gyttja. Decreasing sand content with depth. | High lake levels. One visible and one microscopic tephra layer. High microcharcoal and green algae concentrations. High TOC/TN ratios. High sedimentation rate. |
| | Late Pleistocene | MIS 2 | 6–2.4 | 41.9 | 12.6 | **2.9–2.4 m** – High fine sand (<250 µm), with clayey silt and evidence of lamination. | Increasing lake level. Two cryptotephra layers. Transient nutrient pulse 12.8–11.7 ka. Moderate TOC and low TIC. |
| | | | | | | **6–2.9 m** – Homogenous sediment structure. Silt, distinct lamination and siderite precipitation. | Evidence for ice-rafted debris deposition. Low productivity and lake level. High K and organic δ$^{13}$C. Low water δ$^{18}$O. Declining C/N ratios. High sedimentation rate. |
| | | MIS 3 | 11–6.1 | 32.8 | 7.2 | **6.6–6.1 m** - Massive sediment structure. Silt with distinct lamination. | Steady decrease in lake level. High oxygen index. |
| | | | | | | **11–6.6 m** – Massive sediment structure. Silt. | Increasing lake level. Four visible and three microscopic tephra layers. High C/N ratios. Moderate TOC, very low TIC. |
| | | MIS 4 | 13.9–11 | 33.7 | 9.4 | Massive sediment structure. Clayey silt. | High sedimentation rate. Very low TOC. No tephra layers. Low productivity. Declining C/N ratios. High K content gives evidence for ice-rafted debris deposition. |
| | | MIS 5a-c | 17.7–13.9 | 44.2 | 10.0 | **15.2–13.9 m** - Massive, bioturbated sediments. Clayey silt and fine sand. | Increasing lake level and high productivity. *Dreissena* shell layer 14.58–14.56 m. |
| | | | | | | **17.8–15.2 m** – Massive sediment structure. Clayey silt with fine sand (a). | Deep lake with moderate/low productivity High green algae concentrations. High TOC, low TIC. |
| Lake Ohrid | Holocene | MIS 1 | 4.6 – 1.1 | 47.2 | 26.2 | **3–0 m** – Massive sediment structure. Bright colouring indicates high calcite; dark colouring indicates lower calcite. | High productivity. Four microscopic tephra layers. Low K concentrations. High sedimentation rate. |
| | | | | | | **4.6–3 m** – Slightly calcareous silty clay and massive sediment structure. Frequent siderite-rich layers. | Low TIC and calcite. High iron availability. Low productivity and stronger calcite dissolution. High K concentrations. High sedimentation rate. |
| | Late Pleistocene | MIS 2 | 11.3 – 4.6 | 69.2 | 45.5 | Silty clay. Mottled, often massive sediment structure. Frequent siderite-rich layers. Abundant fine fraction (< 4 µm) sediments. | Very low TIC, TOC, and calcite suggesting low productivity, with large inputs of fine-grained, and chemically weathered siliciclastics. High iron availability. Two visible and two microscopic tephra layers. Mass-movement deposit at 7.87 m. |
| | | MIS 3 | 23–11.3 | 50.6 | 33.4 | | |
| | | MIS 4 | 28.8–23 | 50.2 | 29.6 | | |
| | | MIS 5a-c | 36.3–28.8 | 36 | 20.4 | **35.6 – 28.8 m** – Silty clay with a massive sediment structure. Bright colouring indicates high calcite; dark colouring indicates lower calcite. | Low siliciclastic mineral abundance. Decreasing δ$^{18}$O and δ$^{13}$C. Strong primary productivity. Low sedimentation rate. |
| | | | | | | **36.6 – 35.6 m** – Silty clay. Mottled, often massive sediment structure. Frequent siderite-rich layers. | Higher carbonate δ$^{18}$O and δ$^{13}$C corresponds to reduced TIC, and high siderite. Low sedimentation rate. |

* MIS 5a-c – 96–71 ka; MIS 4 – 71–57 ka; MIS 3 – 57–29 ka; MIS 2 – 29–12 ka; MIS 1 – 12–0 ka.

**Summarised from the following references:

**Lake Prespa** - (Aufgebauer et al., 2012; Cvetkoska et al., 2015; Damaschke et al., 2013; Leng et al., 2013; Panagiotopoulos et al., 2014; Wagner
et al., 2014)
**Lake Ohrid** - (Francke et al., 2016, 2019; Just et al., 2015; Lacey et al., 2016; Leicher et al., 2021; Wagner et al., 2019)

### 4.1. Host Phase Controls

The availability and abundance of specific host phases is often assumed to control sedimentary Hg
accumulation and burial (Outridge et al., 2007). Both Lake Prespa and Lake Ohrid show evidence for
complex relationships between $Hg_T$, TOC, TS, and K concentrations through time (**Fig. 4**). However,
the trends displayed in **Figure 4** also suggest that: (1) the strength of the relationships between Hg,
TOC, TS, and detrital minerals (K) are distinctly different between the two lakes, and (2) the $Hg_T$
signals preserved in Lake Prespa and Lake Ohrid cannot be fully explained by variability in
abundance of these potential host phases individually.

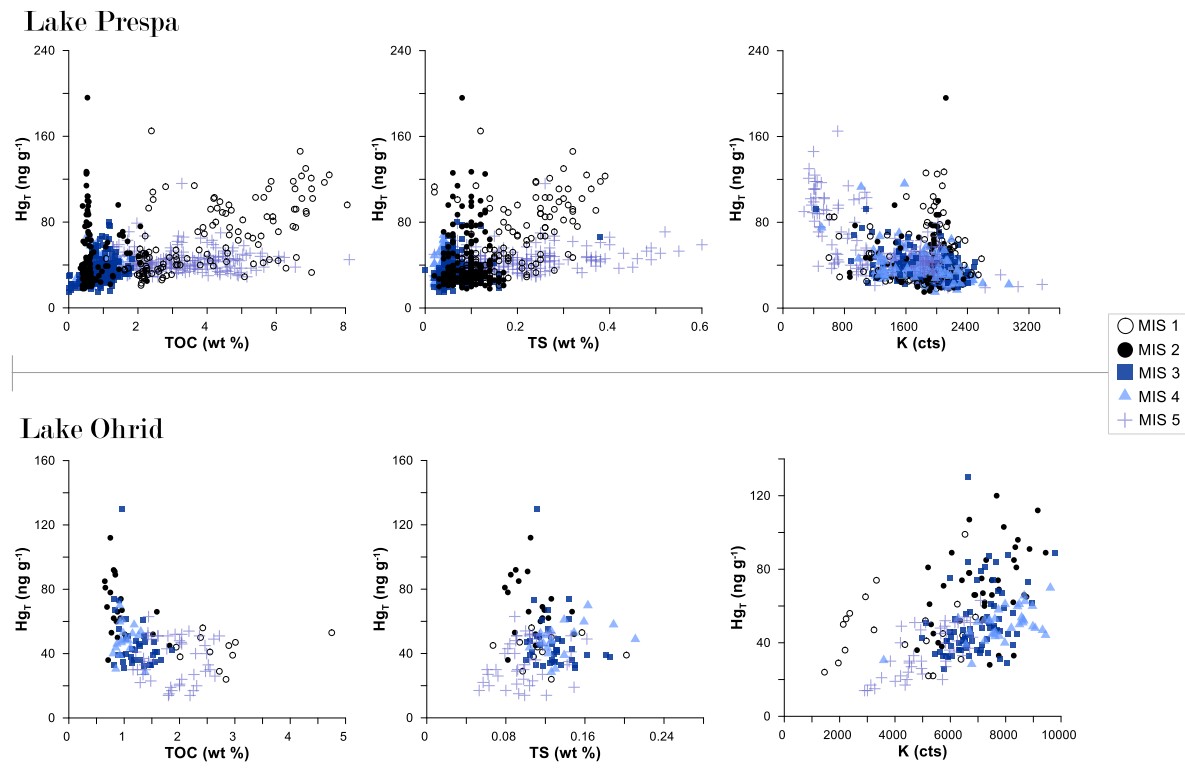

**Figure 4**: A comparison of host-phase relationships between lakes Prespa and Ohrid. Points are coded relative to stratigraphic period: the Holocene (12–0 ka, transparent circles), and the late Pleistocene (90–12 ka, filled symbols). We compare $Hg_T$ records for both lakes relative to total organic carbon (TOC), sulphide (estimated by total sulphur (TS)), and detrital minerals (estimated by potassium (K) concentrations) – note that aluminium (Al) data are more commonly used as an indicator of detrital mineral abundance but these are currently unavailable for 5045-1.


Core Co1215 from Lake Prespa shows a moderate correlation between $Hg_T$ and TOC during the
Holocene and late Pleistocene (all data in **Fig. 4; Table 1**). This correlation is most significant during
the Holocene (MIS 1), where distinct enrichments in $Hg_T$ occur in conjunction with a similarly sharp
increase in TOC, and low variability in Hg/TOC values (**Fig. 5**). However, it is more inconsistent during
the late Pleistocene (MIS 2–5). For example, the highest $Hg_T$ values are measured in the relatively
TOC-lean sediments of MIS 2 (**Fig. 4, 5**), and a plateau also appears when higher TOC
concentrations are reached during MIS 5 whereby $Hg_T$ no longer increased in step with TOC (**Fig. 4,**
**S2**). The correlations observed are not strong enough to conclude that TOC availability can fully
explain the Hg signals observed in Lake Prespa throughout the 90-kyr succession.
Correlations between $Hg_T$, detrital mineral and/or TS availability are also largely absent, suggesting
that the complex Hg/TOC relationship is not a function of time-varying sulphides and detrital mineral
availability. Large peaks in Hg/K are visible during the Holocene (**Fig. 5**), but these are not reflected in
$Hg_{AR}$ and therefore an artefact of considerably lower K concentrations within this section of the core
rather than indicators of changes in lake Hg levels. The highest positive $r^2$ value between $Hg_T$ and TS
is observed during the Holocene (MIS 1: $r^2 = 0.25$) (**Fig. 4**), implying that >75 % of variance in the
dataset cannot be explained with sulphide availability during this time period. Correlations for other
periods are even weaker and some periods appear to show distinct patterns of Hg and potential host-
phase behaviour (**Fig. 4**).
One possibility is that Hg signals reflect changes in the dominant sources of organic and detrital
materials deposited in the lake. For example, combined isotopic and sedimentological data record
episodes of stronger algal blooms during MIS 1 and 5 (Leng et al., 2013), supported by coeval
abundance of freshwater diatom genera such as *Cyclotella* and *Aulacoseira* (Cvetkoska et al., 2015).
All correspond to elevated $Hg_T$, and so could imply more effective Hg burial by autochthonous organic
material compared to allochthonous (Leng et al., 2013; Damaschke et al., 2013). However, in the
presence of abundant binding ligands such as for the Lake Prespa record, maximum Hg burial is
limited principally by supply regardless of productivity, and so changing Hg signals in Lake Prespa
more likely reflect changes in environmental Hg availability; resulting from externally-driven
oscillations in Hg emission and/or exchange between (local) surface reservoirs such as forests, water
courses, and soils (Bishop et al., 2020; Obrist et al., 2018)). This interpretation is supported by the
lack of a close statistical correspondence between Hg, organic matter, sulphur, or detrital mineral
content, source, or composition (**Fig. 4**), which suggests that Hg burial efficiency is only weakly
associated with host phase availability in this system.

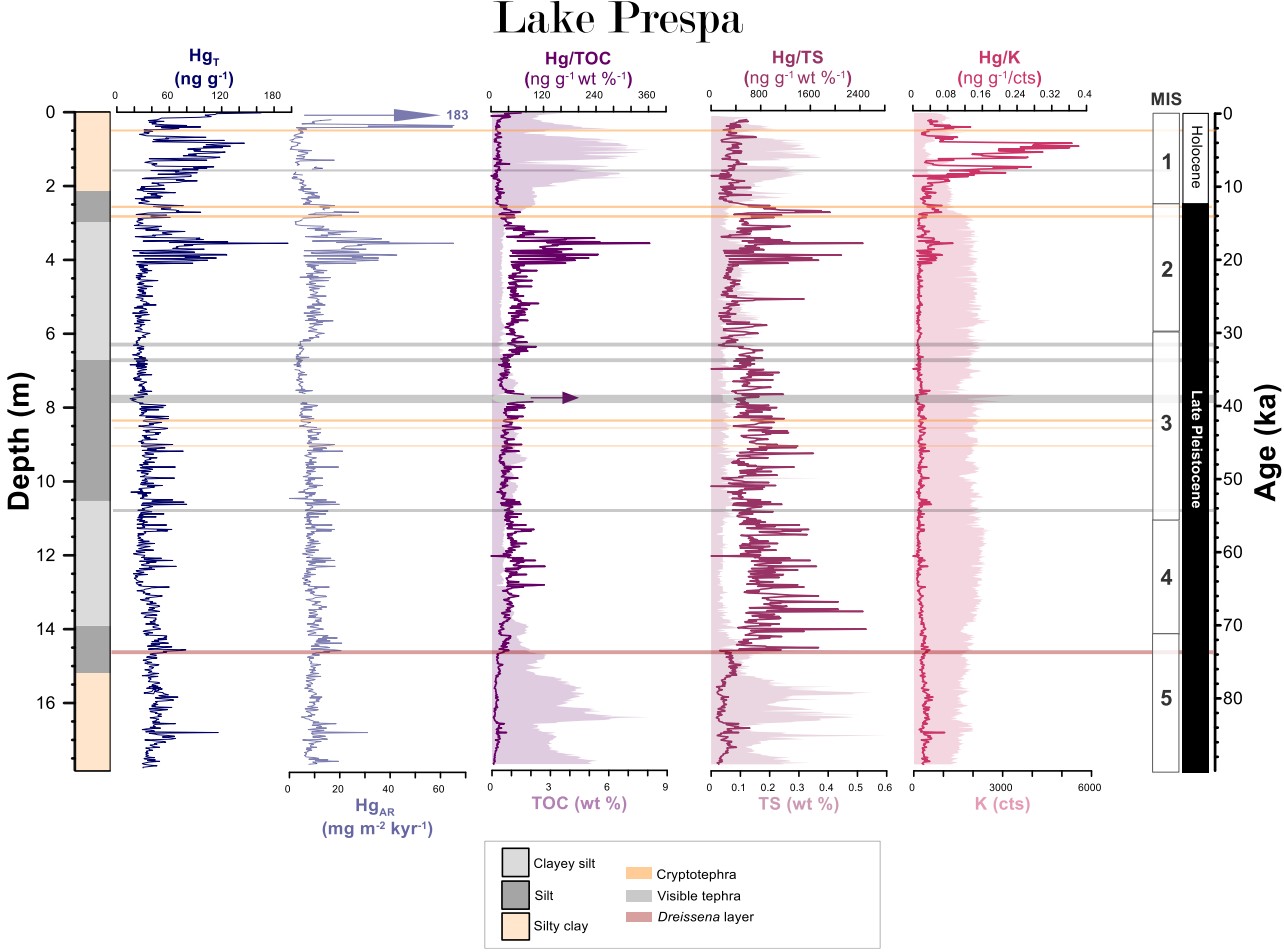

**Figure 5**: Total Hg (Hg$_T$) and total Hg accumulation rate (Hg$_{AR}$) for core Co1215 from Lake Prespa, presented as a function of depth and time, and relative to lithofacies, visible (grey shading) and cryptotephra (orange shading) layers. We include records of Hg$_T$ (this study) normalized to records of total organic carbon (TOC) (Damaschke et al., 2013), total sulphur (TS) (Aufgebauer et al., 2012), and detrital mineral abundance (estimated by potassium (K)) (Panagiotopoulos et al., 2014), with filled shading marking the original datasets. A distinct lake low stand based on seismic profiles and sedimentological data is marked at 14.63 - 14.58 m depth (red shading) (Wagner et al., 2014a). A purple arrow marks sections where artificially high Hg/TOC values are generated by a sharp drop to near-zero TOC (<0.06 wt %) coinciding with deposition of the Y-5 (17.1 m) tephra unit – an effect expected as background sedimentation is interrupted by volcanic ash deposition. White boxes mark the marine isotope stages defined by (Lisiecki and Raymo, 2005), and stratigraphic periods are labelled in black/white.


Core 5045-1 from Lake Ohrid shows elevated Hg$_T$ during the late Pleistocene compared to the
Holocene (**Fig. 6; Table 1**). Peaks in Hg$_T$ most consistently correspond to increases in K (detrital
mineral) intensities, reflected in a broadly positive relationship between Hg$_T$ and K throughout the
succession (**Fig. 4, S3**). However, this relationship is only described by r$^2$ values <0.5 and the
strength of this correlation varies across the span of the record, weakening during the Holocene (**Fig.**
**4**).
Variable Hg values in the Ohrid record appear less influenced by organic matter and/or sulphide
availability. Fluctuations in TOC/TS values suggest that some sulphide formation may have occurred
during the late Pleistocene (MIS 2-5) (Wagner et al., 2009; Francke et al., 2016). However, even in
these phases, TS remains low and correlations between Hg$_T$ and TS are generally negative or weak
(r$^2$ < 0.2; **Fig. 4**) so that Hg signals do not change in magnitude or expression even when TS
variability is accounted for (**Fig. 6**), potentially due to the oligotrophic state of Lake Ohrid favouring
burial of sulphide-depleted sediments (Francke et al., 2016; Vogel et al., 2010). More remarkable, the
relationship between $Hg_T$ and organic matter in Lake Ohrid also shows an inverse correlation (**Fig. 4**).
These trends may be explained by a scenario where the Hg flux to Ohrid from direct deposition and/or
surrounding catchment is typically the limiting factor, rather than availability of potential host phases.

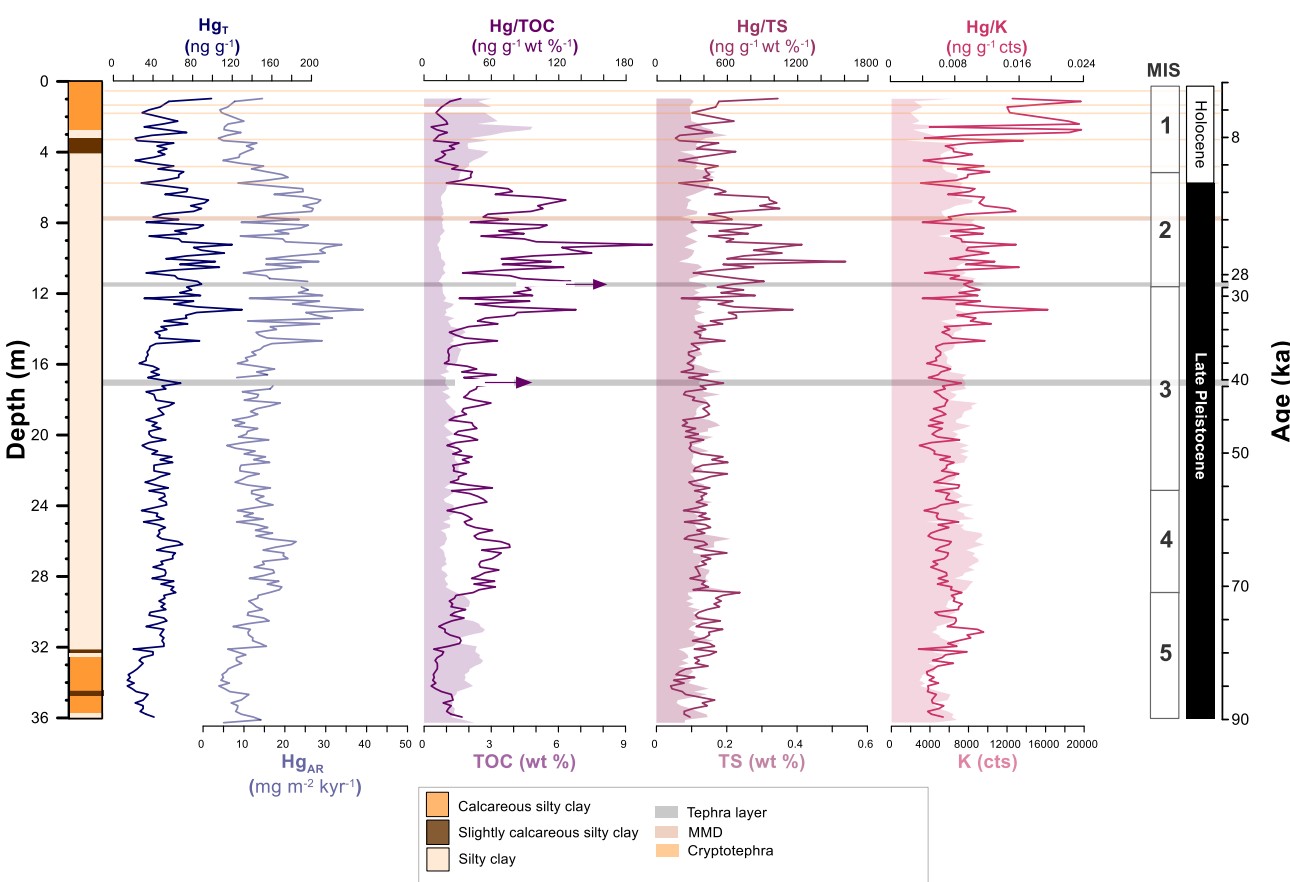

**Figure 6**: Total Hg ($Hg_T$) and total Hg accumulation rate ($Hg_{AR}$) for core 5045-1 from Lake Ohrid, presented as a function of depth and time, and relative to lithofacies, visible (grey shading) and cryptotephra (orange shading) layers. We include records of $Hg_T$ (this study) normalized to records of total organic carbon (TOC) (Francke et al., 2016), sulphide (estimated by total sulphur (TS)) (Francke et al., 2016), and detrital mineral abundance (estimated by potassium (K)) (Francke et al., 2016; Wagner et al., 2019), with filled shading marking the original datasets. A mass movement deposit (MMD) is marked at 7.87 m depth (brown shading) (Francke et al., 2016). Purple arrows mark sections where artificially high Hg/TOC values are generated by a sharp drop to near-zero TOC (<0.06 wt %) coinciding with deposition of the Y-5 (17.1 m) and Mercato (11.5 m) tephra layers – an effect expected as background sedimentation is interrupted by volcanic ash deposition. White boxes mark the marine isotope stages as defined by Lisiecki and Raymo (2005), and stratigraphic periods are labelled in black/white.


Lake Ohrid and Lake Prespa show distinct differences in the strength of their Hg-host phase
relationships. In Lake Prespa, Hg broadly covaries with organic matter (TOC), whereas in Lake Ohrid
correlations are observed between Hg and detrital minerals (K). Nonetheless, only a relatively small
proportion of Hg variability can be explained by host phase availability in each record. This suggests
that while host phase availability may, at times, exert an influence on the Hg signals recorded in these
lakes, the catchment-controlled changes in Hg fluxes are typically the more dominant effect on Hg in
these sediment records. In the absence of a pronounced host-phase influence, retention of a
measurable Hg signal requires that the net influx of Hg into the lake (e.g., surface runoff, wet/dry
deposition) exceeds the amount leaving the system due to processes such as runoff or evasion.
Therefore, we surmise that the $Hg_T$ and $Hg_{AR}$ signals recorded in Lake Prespa and Lake Ohrid are
records of net Hg input to the two lakes rather than the efficiency of sedimentary drawdown.

## 4.2. Tephra layers

As volcanic eruptions are among the most significant natural Hg sources, we assess whether the
previously recognized tephra deposition events in Lake Prespa correspond to changes in Hg
deposition. Overall, we find that individual tephra horizons and surrounding sediments do not
consistently correspond to measurable peaks in $Hg_T$ or $Hg_{AR}$ in Lake Prespa (**Fig. 5**). Only two of the
eleven preserved ash layers coincide with elevated $Hg_T$: Mercato (8.54 ± 0.09 ka; Somma-Vesuvius),
and LN1 (14.75 ± 0.52 ka; Campi Flegrei). These two units are not associated with disproportionately
large tephra volumes and neither coincide with evidence for transient changes in authigenic
carbonate precipitation or sediment diagenesis that may impact sedimentary Hg. This implies that Hg
concentrations in Lake Prespa cannot, in general, be unequivocally linked to short-lived (<1-year)
individual eruption events between ~90 and 0 ka (**Fig. S5**).
Discrete ash fall events (recorded by tephra/cryptotephra) do not consistently correspond to
measurable peaks in $Hg_T$ or $Hg_{AR}$ in the slightly lower-resolution (~400-yr per sample) Lake Ohrid
record (**Fig. S5**). Considering this lack of correspondence of Hg with ash layers, in conjunction with
the Lake Prespa data too, suggests that (a) surface Hg loading was not appreciably increased with
most large eruption events over the past 90 kyr in the Balkans and/or (b) sampling resolution may
need to be significantly higher and/or focused on lesser-bioturbated records to identify single, short-
lived volcanogenic perturbations of the scale and type occurring during the period recorded in the
Ohrid (and Prespa) sedimentary successions.

## 4.3. Variability through time

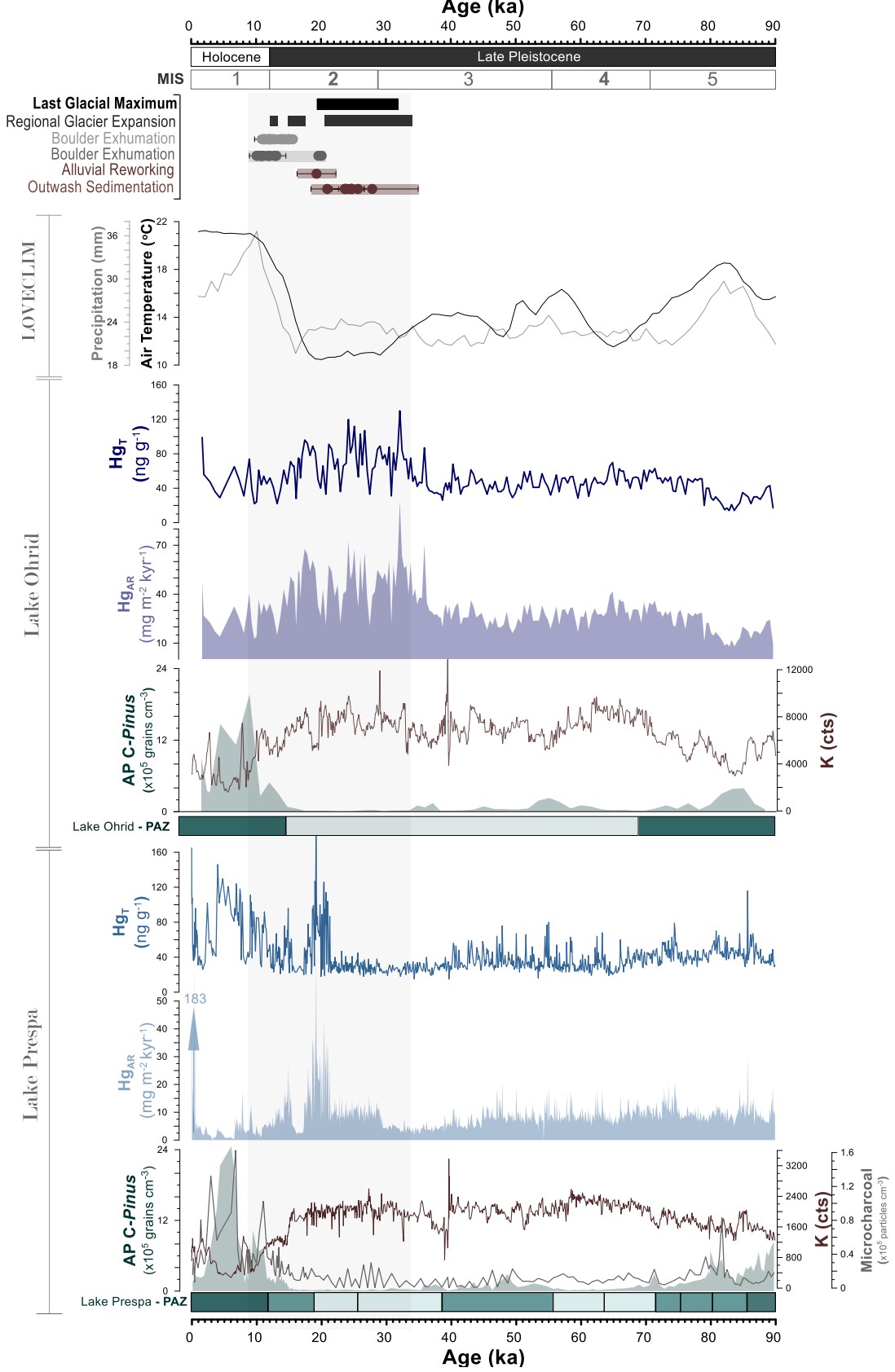

**Figure 7**: Total mercury (Hg$_T$) and mercury accumulation rate (Hg$_{AR}$) records for Lake Prespa and Lake Ohrid generated by this study and proxy datasets generated by prior studies. For Lake Prespa, these include arboreal pollen (AP) concentrations (Panagiotopoulos et al., 2014), microcharcoal (Panagiotopoulos, 2013), potassium (K) (Aufgebauer et al., 2012; Wagner et al., 2010), and pollen assemblage zones (PAZ) (Panagiotopoulos et al., 2014). For Lake Ohrid, these include AP concentrations (Sadori et al., 2016), potassium (K) (Wagner et al., 2019; Francke et al., 2016), 1000-year average surface-air temperature (SAT - °C) and annual mean precipitation (millimetres) both simulated by the LOVECLIM Earth system model (Goosse et al., 2010) for the Prespa/Ohrid region (Wagner et al., 2019), and pollen assemblage zones (PAZ) (Sadori et al., 2016). Pollen assemblage zones defined by Panagiotopoulos et al. (2014) (Lake Prespa) and Sadori et al. (2016) (Lake Ohrid) are presented as green bars, shaded relative to tree population density (darker colour = higher density). We include a chronology of glacial processes based on radiometric dating of glacial landforms in the following locations: the Voidomaitis river basin (purple) (Lewin et al., 1991; Woodward et al., 2008), the Pindus Mountains (lilac) (Allard et al., 2021, 2020; Styllas et al., 2018; Hughes et al., 2006; Pope et al., 2017), and the Dinaric Alps (blue) (Gromig et al., 2018; Ribolini et al., 2018; Ruszkiczay-Rüdiger et al., 2020). White boxes mark the marine isotope stages (MIS) as defined by Lisiecki and Raymo (2005), and stratigraphic periods are labelled in black/white. Vertical grey shading denotes the timing of the largest changes in glacier extent and volume.


## 4.3.1. Late Pleistocene (90 – 35 ka; MIS 5 to MIS3)

The Lake Prespa and Lake Ohrid sediment cores show similarly muted variability in Hg$_T$ and Hg$_{AR}$
values between ~90 and 35 ka (broadly MIS 5a-c, 4 & 3), alluding to relatively stable Hg inputs (**Fig.**
**7; Table 1**). High organic and low clastic material concentrations point to warmer climate conditions
during this interval, in which both catchments experienced an increase in moisture availability,
pronounced forest expansion, and plant diversification – collectively acting to stabilize hillslopes and
reduce deep soil erosion (Francke et al., 2019; Panagiotopoulos et al., 2014; Sadori et al., 2016,
2016). One possibility is that Hg sequestration during this interval was controlled by consistent rates
of algal scavenging (Biester et al., 2018; Outridge et al., 2007, 2019; Stern et al., 2009). Elevated
TOC (**Fig. 5**), hydrogen index, TOC/TN, and biogenic carbonate concentrations between ~90 and 71
ka in both Lake Prespa and Lake Ohrid signal nutrient upwelling and increased allochthonous inputs,
in conjunction with elevated primary productivity. For example, Lake Prespa records green algae
accumulation (Cvetkoska et al., 2016, 2015; Leng et al., 2013; Panagiotopoulos et al., 2014), and
sediments rich in biogenic silica (bSiO$_2$) are also evident in Lake Ohrid (Francke et al., 2016). Slow
changes in lake geochemistry associated with these biological processes are consistent with a steady
Hg$_{AR}$ in both Lake Prespa and Lake Ohrid during this time, and absence of any especially pronounced
changes in Hg$_T$. This could suggest that, for a relatively prolonged period (~96–35 ka), Hg flux to the
two lakes did not change with a magnitude sufficient to cause measurable sedimentary changes, and
processes capable of amplifying differences in sedimentary Hg between Ohrid and Prespa were not
particularly influential.
MIS 3 marks the start of slow divergence between the Hg records of Lake Prespa and Lake Ohrid.
During MIS 3, proxy records suggest that conditions in the Prespa/Ohrid region were milder than MIS
4, but cooler and drier than MIS 5 (**Fig. 7**) (Panagiotopoulos et al., 2014; Sadori et al., 2016; Wagner
et al., 2019). Divergent Hg signals could be linked to two climate-driven processes. First, a reduction
in primary productivity in Lake Prespa signalled by decreasing TOC, hydrogen index, and endogenic
carbonate compared to values observed during MIS 5 (Aufgebauer et al., 2012; Cvetkoska et al.,

2016; Leng et al., 2013). Second is an increase in detrital material flux to both lakes (signalled by elevated K count; **Fig. 7**), due to recession of the surrounding forests and subsequently elevated rates of catchment erosion (Damaschke et al., 2013; Francke et al., 2019; Panagiotopoulos et al., 2014; Sadori et al., 2016). This environmental shift is more likely to favour enhanced Hg mobility in the catchment and burial in a system whereby detrital minerals could either constitute the primary host phase or correlate to $Hg_T$; and so could explain the progressive elevation in $Hg_T$ and $Hg_{AR}$ observed in Lake Ohrid (**Fig. 4**).

## 4.3.2. Last Glaciation (35–12 ka; MIS 3 to MIS2)

The timing, amplitude, and expression of Hg signals captured in Lake Prespa and Lake Ohrid change significantly between ~35 and 12 ka (**Fig. 7**). The largest $Hg_T$ and $Hg_{AR}$ peaks in Lake Ohrid coincide with the Last Glacial Maximum (LGM), and begin at ~35 ka (**Fig. 7**). Synchronous enrichments in K, quartz, and Ti (Francke et al., 2016; Wagner et al., 2019) provide evidence for elevated clastic terrigenous matter inputs and erosion, and are consistent with evidence for a significantly less-vegetated catchment (Donders et al., 2021; Sadori et al., 2016). High clastic fluxes into the lake during the LGM could also relate to meltwater run-off from local mountain glaciers (Ribolini et al., 2011), which would transport large volumes of sediment generated by glacial abrasion, quarrying and plucking (Carrivick and Tweed, 2021; Overeem et al., 2017) into the lake basin. Given that Hg sequestration in Lake Ohrid appears partially related to the abundance of detrital minerals for much of the record (**Fig. 4, 5**), these Hg peaks could relate to local, climate-driven shifts in landscape structure associated with glaciation during MIS 2 (**Fig. 4, 7**).

Alternatively or in addition to these local effects, atmospheric mineral dust concentrations were also up to twenty-times higher during the LGM (Simonsen et al., 2019). Mineral dust may be the most important Hg carrier in ice-cores (Jitaru et al., 2009; Vandal et al., 1993), and studies have shown evidence for notable redistribution of terrestrial Hg during the LGM owing to changes in regional atmospheric dust deposition (de Lacerda et al., 2017; Fadina et al., 2019; Pérez-Rodríguez et al., 2015). However, we see no clear evidence atmospheric dust played a major (direct) role in the local Hg cycle in our data. For example, peaks in elemental ratios typically associated with mineral dust deposits (e.g., Zr/Ti) do not correspond to peaks in $Hg_T$ and/or $Hg_{AR}$ (**Fig. S7**) (Vogel et al., 2010), nor loess-based evidence for elevated aeolian dust fluxes over Central Europe and the Balkans during the last glacial maximum (Újvári et al., 2010; Rousseau et al., 2021). Marine sediment records also do not capture measurable changes in Saharan dust influx to the Ionian and Aegean seas corresponding to pronounced Hg signals in Lake Ohrid (**Fig. S7**) (Ehrmann and Schmiedl, 2021). Therefore, we cannot mechanistically link elevated Hg values during MIS 2 in Lake Ohrid to broad-scale changes in atmospheric dust deposition.

The largest $Hg_T$ and $Hg_{AR}$ peaks in Lake Prespa occur between 21.3 ±1.7 (1σ from the Bayesian age model, see **Fig. 3**) ka and 17.5 ±0.7 ka. These signals do not correspond to a measurable change in host phase availability (**Fig. 5**), so it is unlikely that these peaks reflect changes in TOC, TS, and/or K.

However, they do coincide with deglaciation of the Pindus and Dinaric mountains (**Fig. 7**) (Hughes et
al., 2023). Geomorphological evidence suggests that glaciers were present across the Prespa/Ohrid
region between ~26.5 and 15 ka (Belmecheri et al., 2009; Gromig et al., 2018; Ribolini et al., 2018;
Ruszkiczay-Rüdiger et al., 2020), and indeed that periglacial processes created a landscape
characterized by intense weathering, erosion and sediment transport (Hughes and Woodward, 2017;
Allard et al., 2021). Glacial meltwaters thus likely constituted a major source of water input to Lake
Prespa during the last deglaciation. Glaciers are important sinks for atmospheric Hg deposited by
both dry and wet processes (Durnford and Dastoor, 2011; Zhang et al., 2012), and large quantities of
Hg can accumulate in organic-rich frozen soils (permafrost, Schuster et al., 2018). High proportions of
detrital matter within glacial ice, snow, and organic matter facilitate the effective, long-term (>100s-
1000s of years) retention of atmospheric Hg, meaning that rapid snow/ice melt and permafrost
thawing can produce transient 'pulses' of Hg into lakes without a comparable peak in sediment influx
(Durnford and Dastoor, 2011; Kohler et al., 2022). This is consistent with the abrupt and short-lived
increase in Hg concentration retained in Lake Prespa between 21.3 and 17.5 (±1.7–0.7 (1σ)) ka,
which occurs in the absence of a pronounced change in terrigenous elements (e.g., Ti, Rb) or TS
(**Fig. 5, 7**).
Lakes Ohrid and Prespa show two other striking differences in Hg concentration between 35–12 ka
(**Fig. 7**). First, Lake Prespa does not record a distinct $Hg_T$ or $Hg_{AR}$ signal during the LGM, and second,
Lake Ohrid does not record a distinct $Hg_T$ or $Hg_{AR}$ signal corresponding to deglaciation. Given their
close proximity and environmental similarity, both lakes could be expected to record similar overall
signals if the climate-driven processes influencing $Hg_{AR}$ were broadly similar. One plausible
explanation could be a disproportionately large change in Lake Prespa's total volume compared to
Lake Ohrid. Increased abundance of small *Fragilariaceae* and benthic *Eolimna submuralis* diatom
species point to generally low temperatures and lake levels during MIS 2 (Cvetkoska et al., 2015).
These conditions are also indicated by elevated concentrations of ice-rafted coarse sand and gravel
grains, and further suggest persistent ice formation on the lake surface, likely facilitated by the lake's
shallow depth (Damaschke et al., 2013; Wagner et al., 2010; Vogel et al., 2010). It is possible that the
heightened presence of ice at the peak of glaciation served as a natural barrier between the surface
and the sediments of Lake Prespa, effectively slowing the net flux of Hg into delivery of solutes to the
basin. A simultaneous lack of ice cover on Lake Ohrid, linked to greater water depths, could also
justify why $Hg_{AR}$ remained high in this lake during the LGM, as the Hg influx pathway would be
unaffected by ice formation (**Fig. 7**).
Water volume changes may have also influenced the hydrological connection between lakes Ohrid
and Prespa during deglaciation (Cvetkoska et al., 2016; Jovanovska et al., 2016; Leng et al., 2010).
Tracer experiments and stable isotope (δ18O) analysis suggest that water draining from Lake Prespa
accounts for a significant proportion of Lake Ohrid's water inflow alongside precipitation (Matzinger et
al., 2006; Wagner et al., 2010; Lacey and Jones, 2018), with high rates of prior calcite precipitation
occurring in the connecting karst system (Eftimi et al., 1999; Leng et al., 2010; Matzinger et al., 2006).
However, a change to lower δ18O of lakewater and TIC in both lakes during the last glaciation point to
a reduction in the contribution of karst-fed waters to Lake Ohrid (Lacey et al., 2016; Leng et al., 2013).
Although it is unlikely that the two hydrological systems became completely decoupled (Belmecheri et
al., 2009; Lézine et al., 2010), evidence for permafrost formation at high elevations between 35 and
18 ka (Oliva et al., 2018) and lower precipitation could be linked to a reduction in karst aquifer activity
(**Fig. 7**). For shallower Lake Prespa, lower precipitation may also have led to a larger reduction in lake
volume compared to Lake Ohrid, decrease in the number (and pressure) of active sinkholes, and
subsequently the outflow of water and solutes (e.g., Hg) into Lake Ohrid (Wagner et al., 2014a) –
increasing both $Hg_T$ and $Hg_{AR}$. Together, the collective impact of disproportionately large, climate-
driven reductions in water level could explain why rates of Hg accumulation were significantly higher
in Lake Prespa during deglaciation compared to the LGM. Glacial meltwaters would elevate the net
Hg input compared to the LGM, and reduced ice cover would permit a more direct pathway for Hg to
be delivered into the basin; both processes becoming effective while underground permafrost
continued to limit the intra-basin exchange of water and solutes.
Neither Lake Ohrid nor Lake Prespa show large changes in Hg concentration nor accumulation during
the Oldest (17.5-14.5 ka) and Younger (12.9-11.7 ka) Dryas. Both lakes contain clear evidence for an
abrupt return to glacial conditions during this time. Lake Prespa sediments record shifts in tree pollen
and diatom assemblages alluding to a net reduction in local winter temperatures and moisture
availability (Aufgebauer et al., 2012a; Panagiotopoulos et al., 2013; Cvetkoska et al., 2014), and high
uranium ($^{234}U/^{238}U$) activity ratios, low tree pollen percentages, and low TIC concentrations in Lake
Ohrid also pertain to intense hillslope erosion owing to a more open catchment structure (Francke et
al., 2019b; Lézine et al., 2010). Geomorphological evidence also pertains to local glacier stabilization
(Gromig et al., 2018; Ribolini et al., 2018; Ruszkiczay-Rüdiger et al., 2020) (**Fig. 7**). Nonetheless, we
suggest these events may have been too (a) short-lived, and/or (b) climatically mild to produce a
similarly distinct response in the terrestrial Hg cycle as the processes operating during, and
immediately following, the LGM; potentially explaining the lack of an associated sedimentary Hg
signal.

### 4.3.3. Holocene (12–0 ka; MIS 1)
The timing and amplitude of $Hg_T$ and $Hg_{AR}$ signals recorded in Lake Prespa and Lake Ohrid
sediments are noticeably different during the Holocene (MIS 1). Between 12±0.5 and 3±0.2 ka, Lake
Prespa captures a series of large peaks in $Hg_T$ and $Hg_{AR}$, corresponding to high TOC and TIC
indicative of elevated productivity, higher rates of organic material preservation, and limited mixing
(**Fig. 5**). Conversely, $Hg_T$ and $Hg_{AR}$ show a progressive decline in Lake Ohrid during MIS 1, despite
coeval increases in TOC and TIC (**Fig. 6**). These observations suggest that for most of the Holocene
Hg fluxes into the two lakes were largely decoupled, likely due to differences in catchment and basin
dynamics which impacted the rate of Hg delivery to (and burial in) the lakes.
Divergent Hg signals in Lake Ohrid and Lake Prespa during this time may be linked to heightened
wildfire frequency and/or intensity. Wildfires have the capacity to (in)directly release Hg from

vegetation, and/or through associated changes in soil erosion. Proxy evidence alludes to interglacial conditions characterised by heightened seasonality, characterized by very warm, dry summers coupled with wet, mild winters, an overall increase in the prevalence of deciduous tree species (Cvetkoska et al., 2014; Panagiotopoulos, 2013); but also an increase in macro and microcharcoal concentrations in Lake Prespa (**Fig.7**; Panagiotopoulos et al. 2013). Large wildfires would have a broadly regional-scale impact which, given the close proximity of our two lakes, could theoretically produce a measurable Hg signal in both systems. However, more frequent and/or intense regional fires could also yield measurably different sedimentary Hg signals by their capacity to: (1) enhance surface run off without a corresponding increase in erosion and effectively reduce transport of catchment sourced, mineral-hosted Hg (Mataix-Solera et al., 2011; Shakesby, 2011); (2) enhance downstream transport of Hg released from burned soils and bound to fine and coarse particulate matter (Burke et al., 2010; Takenaka et al., 2021); and/or (3) release large quantities of Hg into the atmosphere following biomass combustion (Howard et al., 2019; Melendez-Perez et al., 2014; Roshan and Biswas, 2023). All three combine to generate impacts that may vary in significance owing to lake-specific differences in sedimentation, accumulation, and flux of materials to/from the lake.

An increase in wildfire activity also corresponds to a period of intensifying human influence in the region; predominantly in the form of land use change, agriculture, and animal husbandry (Cvetkoska et al., 2014; Masi et al., 2018; Panagiotopoulos et al., 2013; Rothacker et al., 2018; Thienemann et al., 2017; Wagner et al., 2009). Widespread mineral resource exploitation and metalworking on the Balkan peninsula is recorded as early as ~8 ka (Gajić-Kvaščev et al., 2012; Longman et al., 2018; Radivojević and Roberts, 2021; Schotsmans et al., 2022), and release of detrital Hg during cinnabar ore extraction and use of Hg in gold extraction (amalgamation) has been linked to pronounced Hg contamination in modern sedimentary units in the region (Covelli et al., 2001; Fitzgerald and Lamborg, 2013). Directly quantifying the influence of (hydro)climate- versus human-driven impacts on sedimentary Hg records presents a major challenge as these factors are interdependent. Nonetheless, these factors could produce a more measurable effect in lake systems with heightened sensitivity to changes in water, nutrient and pollutant fluxes. This could explain why large Hg signals are observed in Lake Prespa between ~12 and 3 ka but not Lake Ohrid: Lake Prespa is shallow relative to its surface area (**Fig. 2**), meaning that relatively small oscillations in pollutant influxes can lead to appreciable changes in lake geochemistry (Cvetkoska et al., 2015; Matzinger et al., 2006).

Decoupling of the two Hg records effectively disappears ~3 ka ago, where both lakes show a sharp and pronounced rise in $Hg_T$ and $Hg_{AR}$ (**Fig. 7**). Several lines of evidence point to human activity as the primary cause. On a local scale, a rapid increase in the biological productivity (eutrophication) of Lake Prespa since ~1.6 (±0.06) ka alludes to greater disturbance of catchment soils by agricultural practices, and eventually use of inorganic compounds such as pesticides and fertilizers (Aufgebauer et al., 2012; Cvetkoska et al., 2014; Krstić et al., 2012; Leng et al., 2013). Signals observed in **Figure 7** may thus be a product of human-induced changes in organic or minerogenic material flux: each facilitating more efficient delivery of catchment-sourced Hg (Fitzgerald et al., 2005), and possibly also stimulating microbial Hg methylation within the sediment (Soerensen et al., 2016). On a broader scale

peaks in $Hg_T$ and $Hg_{AR}$ correspond to a sustained rise in European and/or global Hg emissions, owing to increased deforestation, fossil fuel extraction and combustion, and intentional use of Hg for resource extraction/production (Outridge et al., 2018; United Nations Environment Programme, 2018). An increasing number of sedimentary archives record Hg enrichments as early as ~3 ka ago (Biskaborn et al., 2021; Guédron et al., 2019; Li et al., 2020; Pan et al., 2020). The emergence of simultaneous $Hg_T$ and $Hg_{AR}$ peaks in Lakes Ohrid and Prespa following ~3 ka underscores the magnitude and global distribution of this change in Hg sources and emissions (**Fig. 7**), and point to a rise in Hg fluxes between 3 and 0 ka that was distinct enough to effectively overwhelm previously dominant natural drivers of Hg variability.

## 4.4. Key differences & implications

The magnitude and expression of Hg signals recorded in Lake Prespa and Lake Ohrid are different in three aspects. First, the extent to which different host phases can (or cannot) explain time-varying patterns in Hg concentration differs between the two lakes. Although only a limited fraction of Hg variability in either record can be explained by availability of any single host phase, the low degree of covariance that we do observe points to organic material playing the most significant role as a Hg host in Lake Prespa. In contrast, Hg correlates most strongly with detrital minerals in Lake Ohrid over the same period (0-90 ka) (**Fig. 4**). The second difference is visible during the last glaciation (~35–12 ka): in Lake Ohrid Hg concentrations peak during the LGM (35.8–12 ka), whereas Lake Prespa captures transient, high-amplitude peaks during deglaciation, starting ~15-kyr later (**Fig. 7**). The third difference is visible during the Holocene. The largest signals in the entire Lake Prespa record are observed between ~8 and 0 ka, whereas Hg concentrations do not increase in Lake Ohrid until ~2 ka. These observations raise the question: *for two lakes located in such close geographical proximity and having experienced similar climate conditions, what may have caused such pronounced differences from ~35 ka* (**Fig. 2**)?

Differences in bathymetric structure may offer a plausible explanation. For example, the largest changes in the amplitude and frequency of peaks in $Hg_T$ and $Hg_{AR}$ are exhibited by Lake Prespa (**Fig. 7**): a shallow basin that contains >90 % less water than Lake Ohrid, despite only a ~30 % difference in surface area (Wagner et al., 2010). Increased distance from lake margin to core site in Lake Ohrid would mean distribution of material over a greater total area, and thus more time for net Hg loss to occur either by evasion from the water surface (Cooke et al., 2020), removal of water (and suspended material) via riverine outlets (Bishop et al., 2020), or processes taking place within the water column (Frieling et al., 2023) prior to burial. Therefore, preservation of a measurable Hg signal in a deep lake (e.g., Lake Ohrid) would require notably larger influx of Hg, and this sedimentary signal would also likely be significantly smaller than the equivalent 'dose' delivered to a smaller and/or shallower lake (e.g., Lake Prespa). Coupled with evidence for high-amplitude fluctuations in lake water $\delta^{18}O$ (±6‰) (Leng et al., 2010) and lake level (Cvetkoska et al., 2015, 2016) corresponding to pronounced Hg

variability in Lake Prespa, but not in Lake Ohrid (**Fig. 7**), our data suggest that smaller, shallower
lakes may be particularly sensitive recorders of transient, changes in Hg fluxes.
Divergent bathymetric structures are also linked to distinct differences in biological composition and
nutrient availability in lakes Ohrid and Prespa. The deep (~240 m) waters of Lake Ohrid host a highly
oligotrophic (nutrient poor) environment characterized by low levels of biological productivity, and a
high abundance of planktonic diatom species (e.g., *Cyclotella*) (Cvetkoska et al., 2021). Conversely,
Lake Prespa's shallower (~14 m) waters host a dominantly mesotrophic (nutrient-rich) system in
which benthic and planktonic diatom species are present in equal abundance (Jovanovska et al.,
2016; Cvetkoska et al., 2016), and allude to moderate/high biological productivity (Leng et al., 2013).
Productivity is a potentially important factor influencing the Hg composition of lake sediment: high
productivity typically favours higher concentrations of algal biomass, allowing for more effective Hg
scavenging by organic particles and export to the sediment (Biester et al., 2018; Soerensen et al.,
2016; Hermanns et al., 2013). While the overall signal will remain dominated by Hg availability, broad-
scale differences in productivity between lakes Prespa and Ohrid through time could provide an
additional explanation for the disparate expression of recorded Hg signals (**section 4.1**); with notably
higher productivity in the shallower Lake Prespa further increasing its sensitivity to changes in nutrient
status, erosion, and hydrology.
Local differences in Hg emission by neotectonic activity may have also contributed to the divergent
Hg signals, owing to differences in the host rock geology, tectonic instability, and mechanical stress
regimes of faults surrounding the two basins (Hoffmann et al., 2010; Lindhorst et al., 2015). However,
the significance of these differences cannot be fully assessed in the absence of direct Hg emission
measurements (see **Text SD4**).
The two records presented here highlight that Hg cycling in lacustrine environments is distinct from
open marine systems. In marine systems, Hg fluxes can be broadly modulated by large-scale
continental sediment (Fadina et al., 2019; Figueiredo et al., 2022; Kita et al., 2016) and/or
atmospheric inputs (Chede et al., 2022), and Hg burial flux ultimately becomes more closely related to
host-phase availability. Conversely, both Lake Prespa and Lake Ohrid highlight how the local basin
and catchment characteristics both exert a key control on the delivery of Hg to lacustrine sediments,
and suggest that differences in Hg cycling between geographically-proximal basins could occur as a
function of diverse physical, hydrological, and biological properties.
Our observations highlight that multi-millennial lacustrine Hg records allow a different perspective of
the Hg cycle compared to marine records, and, for example, may be used to infer how local, regional
and global climatic conditions could have altered processes important to the terrestrial Hg cycle.
Because lacustrine records are much better suited to recording smaller-scale processes it is also
clear that extrapolating the (non-marine) Hg cycle response from a single lacustrine Hg record is
challenging. For example, a single-core approach could produce a large degree of uncertainty owing
to variable sediment focussing and catchment-sourced influx of organic and inorganic materials (Blais
and Kalff, 1995; Engstrom and Rose, 2013; Engstrom and Wright, 1984). A valuable next step would

be to apply a source-to-sink approach within a well-known lacustrine catchment: to assess the extent to which Hg sedimentation is spatially heterogeneous within a lacustrine system, and whether multiple cores extracted from different locations within the same basin would yield markedly different Hg trends. Intra-basin heterogenetity in Hg sources, reactions, and transformations could also be examined through measurement of stable Hg isotopes; particularly in millennia-scale sedimentary records where the nature of these processes may change through time (Blum et al., 2014; Jiskra et al., 2022; Kurz et al., 2019). Work of this nature would make great strides toward assessing how representative of variability in the local Hg cycle a single, in this case, lake core is, and whether intra-basin fluctuations in sedimentation, resuspension, and erosion could translate to measurable changes in sedimentary Hg burial.

Past changes in environmental Hg availability inferred from sedimentary records have typically been examined (and presented) by normalizing Hg to a dominant host phase, often taken as organic matter (Fadina et al., 2019; Figueiredo et al., 2020; Grasby et al., 2019; Kita et al., 2016; Percival et al., 2015). However, availability of organic matter or other host phases that scavenge Hg here appear to represent just one of several processes governing Hg burial in lacustrine systems, and this process is very likely systematically less significant compared to marine records in lieu of changes in catchment and basin processes such as erosion, nutrient status, and hydrology (Outridge et al., 2019). Outside pre-industrial times (or periods without an overwhelming global Hg cycle perturbation; such as during LIP formation (Grasby et al., 2019)), a single common process/mechanism is therefore unlikely to produce a unanimous stratigraphic signal across all lakes or even for two adjacent lakes as shown in this study.

# 6. Conclusions

To better understand local and regional impact of climate, vegetation and catchment characteristics on lacustrine Hg records, we present two new high-resolution, Hg records for the last ~90 kyr from Lake Prespa and Lake Ohrid. The two records show some similarities but also distinct differences in the strength of the relationships between Hg, TOC, TS, and detrital minerals (K), with only a relatively small proportion of Hg variability attributable to host phase availability in each record. Our findings provide three valuable insights. First, that local sedimentary environment does influence Hg burial. Covariance with host phases accounts for a limited proportion of the observed variability, suggesting that many of the $Hg_T$ and $Hg_{AR}$ signals recorded in Lake Prespa and Lake Ohrid reflect net Hg input to the two lakes across timescales ranging from decades to multiple millennia. Second, Hg signals can reflect changes in (and also differences between) catchment hydrology and structure. Despite their proximity, the magnitude and expression of the recorded signals are considerably different between Lake Prespa and Lake Ohrid, suggesting these inputs changed relative to sedimentary setting and in response to changing interactions between the two systems. Finally, regional-scale climate variability can measurably affect the Hg signals retained in lake sediments: both lakes Prespa and Ohrid showing changes in Hg concentration and accumulation corresponding to glacial (late Pleistocene)

and interglacial (Holocene) climate conditions. It follows that local, regional, or global changes in
climate or hydrological cycling capable of affecting mineral soils, (peri-)glacial features or fire regime
in the lake catchment could all impact Hg fluxes. These findings prompt further examination of how
orbital-scale climate variability (>$10^3$-year timescales) may influence the terrestrial Hg cycle, not only
to better resolve processes acting on single lacustrine and terrestrial successions, but also to identify
which of these (local) processes could hold relevance for Hg cycling on a global scale.

## Competing Interests

The corresponding author declares that none of the authors have any competing interests.

## Acknowledgements

ARP, IMF, JF, and TAM acknowledge funding from European Research Council Consolidator Grant
V-ECHO (ERC-2018-COG-818717-V-ECHO). ARP thanks Professor David Thomas and Mona
Edwards (School of Geography, Oxford) for logistical assistance with sample transfer and storage. KP
acknowledges funding from the German Research Foundation (DFG grant PA 2664/4-1). All authors
thank members of the Scientific Collaboration on Past Speciation Conditions in Lake Ohrid
(SCOPSCO), and the CRC 806 "*Our Way to Europe - Culture-Environment Interaction and Human*
*Mobility in the Late Quaternary*" projects: for their efforts in producing the Lake Ohrid and Lake
Prespa sediment successions, and making the data available for scientific use.

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
