# Peer review of "Mercury records covering the past 90 kyr from lakes Prespa and Ohrid, SE Europe"

_EGUsphere, 2023_

## Referee Comment (RC2)

This study examines long-term Hg records in sedimentary archives due to its sensitivity to centennial to millennial-scale environmental variations. Sediment analysis from two interconnected lakes, Lake Prespa and Lake Ohrid, over the past 90,000 years reveals distinct Hg patterns. Divergent Hg signals during the early and middle Holocene suggest that local factors significantly influence the Hg cycle's response to environmental changes, highlighting the role of sediment composition, lake structure, and water balance in determining the local versus global influences on Hg signals. It is a very interesting topic.

This paper contains dense content and thorough analysis with well-written explanations. I am curious about whether the biota species are the same in both lakes, as this could be another factor impacting the differences in Hg records between these two bodies of water. Additionally, the layout of the paper could be improved, such as placing tables and figures at the end of the manuscript, which would enhance its readability and organization.

Introduction

1. Line 35-44 In the first paragraph, I believe it would be good to emphasize the bi-directional pathway of Hg transportation. Hg can not only be emitted/released into the atmosphere but can also deposit into terrestrial and oceanic ecosystems.
2. Line 83, please remove this subtitle as there are no other subtitles in the Introduction section.
3. Line 112, it would be good to include the full name of $Hg_{AR}$, as this is the first instance of its mention in the manuscript.
4. Fig 2. Study map normally locates in "2. Site Description".

Site Description

5. Line 148 to 149, delete the dashed line.
6. Line 259 to 265, I recommend merging and simplifying this content with the information found between lines 207 to 211 and lines 215 and 217.
7. Line 260 and 261, It appears that the same method is used to calculate TOC, but there are different references compared to lines 210 and 211. Is there a specific reason for this discrepancy?

Section 3.3 Mercury measurements

8. Line 292 and 293, why use different resolution to analyze Hg sediment samples from these two lakes?
9. Line 293 what is the size of the powered samples, homogenize of sediment samples are really important.
10. Line 296 Could you provide information relating to percent recovery for the standard material?
11. Line 299 Could you please specify the exact table or figure that indicates the calibration results here?
12. Line 303 I recommend removing the subtitle 3.3.1 since there are no other subtitles in this section.
13. Fig 4. The legend for MIS 3-5 in the figure is not easy to identify. It's up to your discretion whether to consider using different colors to improve clarity
14. Line 390, what is p-value for the relationship between $Hg_T$ and TS for MIS 1?

15. I am wondering if biota species are the same between these two lakes Hg pool/accumulation, you have compared the hydrology, sedimentation regime, and geochemistry of them.

---

## Author Comment (AC1)

Please find below our detailed responses (in blue) to comments given by Reviewer #1, where the original reviewer comments are repeated here in black for clarity and completeness.

**Reviewer #1**

https://doi.org/10.5194/egusphere-2023-1363-RC1

Paine et al. presents sedimentary Hg signals and associated processes from two lakes over the past 90ka. Overall, the data quality is high, and the manuscript is well written. It is interesting to see such different Hg signals at the same periods between two lakes just ca. 10km apart. I would like to provide some comments or seek for clarifications.

> We thank Reviewer #1 for taking the time to provide feedback on our manuscript, and for their kind words regarding its presentation and quality. In the response below and in our revised manuscript, we have sought to ensure the critiques raised are suitably considered and addressed where necessary.

**Abstract:**

Line 30. I would suggest adding "Hg sources" since different Hg sources also influence the local Hg cycles as discussed in the manuscript.

> Good point. We will alter this sentence to read:
>
> **Lines 29 – 31***: The lack of coherence in Hg accumulation between the two lakes suggests that, in the absence of an exceptional perturbation, local differences in sediment composition, lake structure**, Hg sources**, and water balance all influence the local Hg cycle…*

**Introduction:**

The introduction section is nice! Paine et al. showed a clear summary diagram on different processes liberating/ mobilizing/ depositing Hg (i.e. Figure 1). I would still suggest adding a few lines on the sources of Hg to the lake in lines 45-53 or in other suitable paragraphs, to help readers better understand why Hg signals could be so different in lakes Prespa and Ohrid.

> We thank reviewer #1 for their kind words. We agree that this (highly relevant) contextual information could be more clearly presented in the introduction, and so will address this through revision of the following sentences:
>
> **Lines 35 - 36***: Mercury (Hg) is a volatile metal released into the atmosphere, lakes, and ocean from both natural and anthropogenic sources, and actively cycled between surface reservoirs*
>
> **Lines 59 - 62:** *Time-resolved sediment records sourced from marine and lacustrine basins are highly suitable for assessing these roles further back in time, as the Hg recorded may originate from one of several potential sources in the atmospheric (e.g., precipitation, dust), terrestrial (e.g., soils, detrital matter), aquatic, and/or lithospheric domain (**Fig. 1**).*
>
> Description of Hg sources is kept general in this section of the manuscript, with the purpose of equipping the reader with a broad overview of the state-of-the-art; prior to targeted discussion of lakes Prespa and Ohrid throughout the rest of the manuscript.

Line 86. How about changes in catchment? I would suggest adding a phrase about catchment in this sentence.

> This is an interesting suggestion, and we will add a statement including this suggestion to the previous paragraph (**lines 57 - 66**) to highlight the role local-regional scale processes may play, particularly for lacustrine records and refer directly to changes in the catchment. For the paragraph here we prefer to retain focus on the soft-sedimentary processes internal to the

lake that could influence Hg composition. To further clarify this, we will revise the following sentences:

*Lines 89 – 92: However, internal changes in bioproductivity, organic matter type and/or flux, sedimentation rate, pH, and redox conditions could all produce a distinct, local, transient, sedimentary Hg enrichment without a meaningful change in the total amount of Hg present and/or mobile in the system.*

*Lines 95 – 97: This impact is then removed, via normalization (e.g., Hg/TOC, Hg/TS), to reveal broader changes in environment Hg availability (Grasby et al., 2019; Percival et al., 2015; Shen et al., 2020; Them et al., 2019).*

*Lines 97 – 99: Such an approach is particularly beneficial for studies typically spanning >$10^2$-year timescales, where the goal is to isolate the effects of catchment-scale depositional and/or transport processes on Hg signals recorded in the sediment through time.*

**Site Description:**

Lines 143-146. Similar to the question above (i.e. line 86), how would changes of vegetation distribution in the catchment influence the Hg input to the lake sediment over different periods? Prespa lakes have a catchment of ca. 1300 $km^2$. I assume there is a significant Hg contribution from the catchment (e.g., through precipitation and then runoff). If not, please clarify.

This is a good question, and one that draws on a critical component of this study. The topic of catchment influences on the Hg signals recorded in the lake(s) is introduced in **section 1**, and then considered with direct reference to lakes Prespa and Ohrid in **section 4.4** of this manuscript. To define this structure more explicitly, we will add the following statement to the beginning of **section 4**:

*Lines 373 – 376: We first consider the extent to which soft sediment processes (**section 4.1**) and lithological features (**section 4.2**.) may have influenced the Hg variability observed in **Figures 5** and **6**, before adopting a catchment-scale perspective in **section 4.3** to explore the role of diverse environmental processes in Hg cycling through these two systems.*

Lines 170. Does runoff from catchment belong to the category of "direct precipitation (35%)"?

No, catchment-sourced run-off falls under the category of 'river runoff'. But we do agree this requires clarification, and so will adjust the wording of the preceding sentence as follows:

*Lines 174 – 175: Water input is sourced from surface runoff (56%), direct precipitation (35%), and inflow from the smaller of the two lakes.*

**Chronology**:

Lines 220-239. The chronologies are very well established! But I would still suggest briefly mentioning the analytical methods of $^{14}$C, $^{40}$Ar/$^{39}$Ar, and ESR in this section or in the supplementary materials, even though some relevant references are already cited. This can provide some pedagogical information to readers on age reconstruction using different techniques.

We agree that full disclosure of chronological details is important in this study. However, full description of these details in the main text may add unnecessary 'bulk' to the manuscript. We will follow the reviewer's suggestion to include a bit more detail on this in the SI text, and two short statements will be added to refer the reader to these methods in the SI:

*Lines 242 – 243: All twenty-seven tie-points and accompanying chronological details are presented in **Text SI3** and **Table S3***

*Lines 293 – 294: Full description of the 5045-1 chronology and associated methods are presented in **Supplementary Text SI4.***

We will also make the following additions to the supplementary file:

(1) Three new sub-sections to our accompanying supplementary file:

**SI3.1. Existing data & methods**

**SI3.2. A new Co1215 chronology**

**SI4.1. Existing data & methods**

(2) Two new tables to our accompanying supplementary file, which present important information related to ESR dating of the *Dreissena* shell layer in core Co1215 (Prespa) as first presented in (Damaschke et al., 2013):

**Table S1**: Radionuclide contents of bulk sediment samples presented in (Damaschke et al., 2013). All analysis was done by high-resolution gamma-ray spectrometry with samples K-5835 and K-5836 (2011) measured at the Cologne lab, and sample K-5800 (2009) analysed at the VKTA lab, Dresden. All errors represent the 1σ level. For dose rate calculation the mean values were used.

| Lab Code | U (ppm) | Th (ppm) | K (%) |
|----------|---------|----------|-------|
| K-5835 | 3.59 ± 0.19 | 17.57 ± 1.02 | 2.37 ± 0.09 |
| K-5836 | 3.53 ± 0.19 | 17.99 ± 1.03 | 2.38 ± 0.09 |
| K-5800 | 3.80 ± 0.40 | 17.30 ± 0.60 | 2.30 ± 0.07 |

**Table S2:** Parameters for dose rate calculation, total dose rates, equivalent dose values and ESR ages presented in (Damaschke et al., 2013). All errors represent the 1σ level.

| Sample | Depth (m) | U (ppm) [shells, ICP-MS] | Dose Rate (Gy kyr$^{-1*}$) | Equivalent Dose (Gy) | ESR Age (ka) |
|--------|-----------|---------|-----------|---------------------|--------------|
| K-5835 | 14.70 – 14.88 | 0.06 ± 0.01 | 1.36 ± 0.10 | 93.71 ± 2.03 | 68.9 ± 5.1 |
| K-5836 | 14.58 – 14.70 | 0.06 ± 0.01 | 1.36 ± 0.10 | 114.41 ± 6.73 | 84.1 ± 7.8 |
| K-5800 | 14.58 – 14.63 | 0.08 ± 0.01 | 1.36 ± 0.10 | 100.2 ± 11.2 | 73.9 ± 9.9 |

* Calculation of dose rates includes the following parameters and assumptions: alpha-efficiency value 0.10±0.02, thickness before and after surface etching 0.82/0.75 mm, an average water content of 47±4.7% (weight water/wet sediment), calculation of the cosmic dose contribution is based on the actual sampling depth including additional shielding through a water column.

The chronologies for both cores also do integrate age estimations for discrete, often independently dated tephra layers. In **Table S6,** we already provide references for the ages given for each of these layers, to ensure the reader can easily access the source publications and their corresponding associated analytical set-ups. This data is also provided in an accompanying spreadsheet.

**Mercury Accumulation:**

Lines 316-318. Are the methods to calculate sedimentation rates and dry bulk density for Lake Ohrid the same as the ones for Lake Prespa?

**Sedimentation rate (SR**) → Yes, the only difference was that two different research teams did these calculations. For Lake Prespa, SR was calculated and presented as mm yr$^{-1}$, whereas for Lake Ohrid SR values were presented as cm yr$^{-1}$. This meant that different calculations were required to convert these values to m kyr$^{-1}$ for use in this study.

**Dry bulk density (DBD)** → No, but only because analyses were conducted by different studies/research teams. For Lake Ohrid, density data were directly measured by Francke et al. (2016), whereas for Lake Prespa we calculated DBD using TOC and water content data obtained by Aufgebauer et al. (2012) and Damaschke et al. (2013). For our calculations, we assumed an average wet density of 1 g cm$^{-3}$ for wet sediments, and 2.6 g cm$^{-3}$ for dry sediments.

To clarify these methodological details in the manuscript text, we will make the following addition to section 3.4:

> **Lines 323 – 324**: *Sedimentation rate (SR) values for both Prespa and Ohrid were calculated by combining stratigraphic and lithological observations with the age-depth relationship ascertained for each core, respectively.*
>
> **Lines 326 – 328**: *DBD values were calculated on the basis of sedimentological data available for each core. For the Lake Ohrid dataset, DBD values were already available following the analyses of Francke et al. (2016). To acquire these values for Lake Prespa, we employed the formula…*
>
> **Lines 332 – 333**: *…assuming an average wet density of 1 g cm$^{-3}$ for wet sediments, and 2.6 g cm$^{-3}$ for dry sediments.*

**Results and Discussion:**

Lines 380-381 and Figure 4. Hg$_T$ concentration appears consistently high during MIS2 in both Lake Prespa and Ohrid. Where does the Hg come from?

Good observation, and one we highlight as a key defining feature of the presented records. This section focusses solely on the soft-sediment processes that may be influencing the Hg concentration of the two records; after the Hg has reached the lake. So, rather than changing the wording of this sentence to introduce different Hg sources (and potentially compromise overall clarity), in our revised manuscript we will better guide the reader through this narrative flow by adding the following statement to **section 4**:

> **Lines 373 – 376**: *We first consider the extent to which soft sediment processes (**section 4.1**) and lithological features (**section 4.2**.) may have influenced the Hg variability observed in Figures 5 and 6, before adopting a catchment-scale perspective in **Section 4.3** to explore the role of diverse environmental sources and processes in Hg cycling through these two systems.*

Lines 393-397. This explanation is quite superficial, even though it makes sense. I would suggest going deeper to find evidence to explain it a bit more. For example, (1) how did the catchment shift regarding vegetation? (2) what can shift the rates of Hg emissions and/or exchange between surface reservoirs? Hg loss by reduction of Hg$^{2+}$ in lake ecosystems can be very important (e.g., by photoreduction, Jiskra et al., 2021. https://doi.org/10.1021/acsearthspacechem.1c00304 )

Omission of information/discussion related to catchment-scale processes from this section is deliberately done in order to maintain a clear narrative structure. We hope that the inclusion of the statement referenced above clarifies this reasoning, which we agree could have been more clearly provided in our original submission. The reviewer also raises some very valid points regarding the strength of our interpretation as written, and we agree that this could benefit from additional supporting evidence. In light of this comment, we will edit the passage in focus as follows:

> **Lines 416 – 429**: *… On one hand, Hg signals could reflect changes in the dominant sources of organic and detrital materials deposited in the lake. For example, combined isotopic and sedimentological data record episodes of stronger algal blooms during MIS 1 and 5 (Leng et al., 2013), supported by coeval abundance of freshwater diatom genera such as Cyclotella and Aulacoseira (Cvetkoska et al., 2015). All correspond to elevated Hg$_T$, and so could imply more effective Hg burial by autochthonous organic material compared to allochthonous (Leng et al., 2013; Damaschke et al., 2013). However, in the presence of abundant availability of binding ligands such as for the Lake Prespa record, maximum Hg burial is limited principally by supply regardless of productivity, and so changing Hg signals in Lake Prespa more likely reflect changes in environmental Hg availability; resulting from externally-driven oscillations in Hg emission and/or exchange between (local) surface reservoirs such as forests, water courses, and soils (Bishop et al., 2020; Obrist et al., 2018)). This interpretation is supported by the lack of a close statistical correspondence between Hg, organic matter, sulphur, or detrital mineral content, source, or composition (**Fig. 4**), which suggests that Hg burial efficiency is only weakly associated with host phase availability in this system.*

Regarding point (1), the influence of vegetation changes will be discussed in greater detail in section 4.4. For example:

> **Lines 489 – 493**: *High organic and low clastic material concentrations point to warmer climate conditions during this interval, in which both catchments experienced an increase in moisture availability, pronounced*

*forest expansion, and plant diversification – collectively acting to stabilize hillslopes and reduce deep soil erosion (Francke et al., 2019; Panagiotopoulos et al., 2014; Sadori et al., 2016, 2016).*

Concerning point (2), we agree that evasion also should have been given more sufficient credence in our original submission. In light of this suggestion, we will integrate more explicit mention of this process at several points in our revised manuscript. For example:

**Lines 50 – 53**: *Evasion back to the atmosphere, consumption by living organisms, or sequestration within aquatic sediment all represent ways in which Hg may 'leave' the terrestrial environment; the latter are known to be particularly effective sinks within the global Hg cycle (Bishop et al., 2020; Selin, 2009).*

**Lines 457 – 461**: *In the absence of a pronounced host-phase influence, retention of a measurable Hg signal requires that the net influx of Hg into the lake (e.g., surface runoff, wet/dry deposition) exceeds the amount leaving the system due to processes such as runoff or evasion. Therefore, we surmise that the $Hg_T$ and $Hg_{AR}$ signals recorded in Lake Prespa and Lake Ohrid are records of net Hg input to the two lakes rather than the efficiency of sedimentary drawdown.*

It is possible that photoreduction did influence the Hg composition of the Prespa and/or Ohrid sediments. For example, the photoreduction of divalent Hg to gaseous Hg (more susceptible to evasion) is controlled mainly by dissolved organic matter (DOM) in aquatic settings (Luo et al., 2017; O'Driscoll et al., 2018), and several Holocene-age lake and peat cores show evidence for increased evasion of Hg corresponding to sediments enriched in organic matter and/or containing evidence for heightened productivity. Collectively, this could point to more effective Hg photoreduction under warm climatic conditions (Jiskra et al., 2015; Schaefer et al., 2020; Biester et al., 2018; Hermanns et al., 2013), but potentially also more in Lake Prespa where Hg variability is more closely correlated to TOC variability. However, we are limited in our ability to constrain this flux for either of the two lakes, and further speculation on climate-driven differences in Hg loss by photoreduction may not be a valuable/useful addition to our revised manuscript:

(1) **The extent to which photoreduction has influenced the Hg composition of a sediment record is best distinguished through use of stable isotopes, which were not measured in this study**. Isotope analysis can provide crucial indications of whether photoreduction of divalent Hg to gaseous Hg has occurred, most commonly by identification of odd-mass Hg stable isotope anomalies (e.g., $\Delta^{199}$ and $\Delta^{201}$ Hg), (Kurz et al., 2019; Jiskra et al., 2022). Given this data currently are not available for lakes Prespa and Ohrid, we cannot confidently distinguish the mechanisms by which Hg is removed from these systems beyond simple speculation; nor the extent to which these mechanisms influenced the observed Hg composition.

(2) **Studies in which isotopic methods have been applied are nearly all limited to the Holocene (<11 ka)**. The most robust estimates of terrestrial Hg evasion are mainly based on enriched isotope tracing studies of recently (<$10^2$-year) deposited sediments (Obrist et al., 2018); many of which are focused primarily on the marine environment (Soerensen et al., 2016; Jiskra et al., 2021; Horowitz et al., 2017). This means that there are currently few studies that seek to quantify the influence that climate shifts associated with glacial-interglacial conditions may have exerted on Hg evasion and/or photoreduction. Even fewer explore this in the terrestrial realm; limiting the applicability of present understanding to the records presented here.

There remains much to learn about the ways in which Hg loss may affect the signals retained in lake sediments on multi-millennial timescales. However, despite the aforementioned uncertainties, we do fully concur with reviewer #1's point that this component of the Hg cycle is important and well worth further study; especially in older, pre-Holocene records such as those from lakes Prespa and Ohrid. Directly inspired by this comment, we will also add reference to the potential of Hg isotope analysis in this research domain in the closing **section 4.4**:

**Lines 743 – 746**: *Intra-basin heterogeneity in Hg sources, reactions, and transformations could also be examined through measurement of stable Hg isotopes; particularly in millennia-scale sedimentary records*

*where the nature of these processes may change through time (Blum et al., 2014; Jiskra et al., 2022; Kurz et al., 2019).*

Lines 535-554. It is not clear to me why Hg accumulation profile in Lake Prespa spanning 10 ka from 33 to 23 ka is much flatter than the one in Lake Ohrid. Why isn't Hg accumulation elevated in Lake Prespa as the one in Lake Ohrid during this period? Does it link to the shallow characteristic of Lake Prespa or limited Hg input?

We have split out this comment to answer the two specific queries (see also below). This certainly warrants further clarification both here and in our revised manuscript. At the core of our interpretation are the differences that exist between the two lakes in terms of bathymetric structure, catchment characteristics, and the extent to which both of these were affected by glaciation. Independent evidence has shown Lake Prespa to be significantly more sensitive to transient climate change than Lake Ohrid (Jovanovska et al., 2016), and so **section 4.3** seeks to explore how this differing sensitivity could also influence Hg cycling. Reviewer #1 rightly points out that we have omitted to mention a key (and intriguing) feature of the two profiles: the distinct lack of a $Hg_{AR}$ signal during the LGM in Lake Prespa. First, we will include a more direct reference to this feature:

**Lines 566 – 570**: *Lakes Ohrid and Prespa show another two striking differences in Hg composition between 35–12 ka (**Fig. 7**). First, Lake Prespa does not record a distinct $Hg_T$ or $Hg_{AR}$ signal during the LGM, and second, Lake Ohrid does not record a distinct $Hg_T$ or $Hg_{AR}$ signal corresponding to deglaciation. Given their close proximity and environmental similarity, both lakes could be expected to record similar overall signals if the climate-driven processes influencing $Hg_{AR}$ were broadly similar.*

This will be followed by a more explicit discussion of the potential reasons underpinning the observed signals; discussion directly inspired by the questions raised in this review:

Why isn't Hg accumulation elevated in Lake Prespa as the one in Lake Ohrid during this period?

**Lines 570 – 580**: *One plausible explanation could be a disproportionately large change in Lake Prespa's total volume compared to Lake Ohrid, with implications for seasonal ice cover…It is possible that the heightened presence of ice at the peak of glaciation served as a natural barrier between the surface and the sediments of Lake Prespa, effectively slowing the net flux of Hg into delivery of solutes to the basin. A simultaneous lack of ice cover on Lake Ohrid could also justify why $Hg_{AR}$ remained high in this lake during the LGM, as the Hg influx pathway would be unaffected by ice formation (**Fig. 7**).*

**Lines 581 – 594**: *Volume changes may have also influenced the hydrological connection between lakes Ohrid and Prespa during deglaciation (Cvetkoska et al., 2016; Jovanovska et al., 2016; Leng et al., 2010)…For Lake Prespa, a measurable change in lake volume would reduce the number (and pressure) of active sinkholes, and subsequently the outflow of water and solutes (e.g., Hg) into Lake Ohrid(Wagner et al., 2014) – increasing both $Hg_T$ and $Hg_{AR}$.*

Does it link to the shallow characteristic of Lake Prespa or limited Hg input?

**Lines 594 – 600**: *Together, the collective impact of disproportionately large, climate-driven reductions in water level could explain why rates of Hg accumulation were significantly higher in Lake Prespa during deglaciation compared to the LGM. Glacial meltwaters would elevate the net Hg input compared to the LGM, and reduced ice cover would permit a more direct pathway for Hg to be delivered into the basin; both processes becoming effective while underground permafrost continued to limit the intra-basin exchange of water and solutes.*

**Key differences and implications**

The whole section is overall well written, but it is lack of some interpretation on Hg loss from my perspective. Hg loss can be very different between these two lakes, therefore affecting the net Hg signals in the sediments. I would suggest adding a few lines on this information to make your interpretation more convincing.

We thank the reviewer for their kind critique, and equally for giving constructive pointers from which to improve this section of the manuscript. As discussed above, we are cautious to put precise constraints on the extent to which different processes may have influenced Hg loss from the two lakes (e.g., evasion, photoreduction). Nonetheless, we concur that Hg loss is a vital component of lacustrine Hg cycling that warrants mention in this section, and so we will make the following text additions:

**Lines 692 – 697:** *Increased distance from lake margin to core site means distribution of material over a greater total area, and thus increased potential for net Hg loss either by evasion from the water surface (Cooke et al., 2020), removal of water (and suspended material) via riverine outlets (Bishop et al., 2020), or processes taking place within the water column (Frieling et al., 2023). Therefore, preservation of a measurable Hg signal in a deep lake (e.g., Lake Ohrid) would require a sufficiently large influx of Hg…*

This will add to the interpretations made in the preceding section (**4.4**), where we already speculate on the potential for loss of Hg from Lake Prespa as a result of transport through the karst system underlying the two lakes:

**Lines 587 – 589:** *…decreases in the reconstructed δ¹⁸O of lake water and TIC in both lakes during the last glaciation point to a reduction in the contribution of karst-fed waters to Lake Ohrid (Lacey et al., 2016; Leng et al., 2013).*

**Lines 592 – 594:** *For Lake Prespa, a measurable change in lake volume would reduce the number (and pressure) of active sinkholes, and subsequently the outflow of water and solutes (e.g., Hg) into Lake Ohrid (Wagner et al., 2014) – increasing both $Hg_T$ and $Hg_{AR}$.*

**References Cited**

Biester, H., Pérez-Rodríguez, M., Gilfedder, B.-S., Martínez Cortizas, A., and Hermanns, Y.-M.: Solar irradiance and primary productivity controlled mercury accumulation in sediments of a remote lake in the Southern Hemisphere during the past 4000 years: Primary productivity and mercury accumulation, Limnol. Oceanogr., 63, 540–549, https://doi.org/10.1002/lno.10647, 2018.

Bishop, K., Shanley, J. B., Riscassi, A., de Wit, H. A., Eklöf, K., Meng, B., Mitchell, C., Osterwalder, S., Schuster, P. F., Webster, J., and Zhu, W.: Recent advances in understanding and measurement of mercury in the environment: Terrestrial Hg cycling, Science of the Total Environment, 721, https://doi.org/10.1016/j.scitotenv.2020.137647, 2020.

Cvetkoska, A., Jovanovska, E., Francke, A., Tofilovska, S., Vogel, H., Levkov, Z., Donders, T. H., Wagner, B., and Wagner-Cremer, F.: Ecosystem regimes and responses in a coupled ancient lake system from MIS 5b to present: the diatom record of lakes Ohrid and Prespa, Biogeosciences, 13, 3147–3162, https://doi.org/10.5194/bg-13-3147-2016, 2016.

Damaschke, M., Sulpizio, R., Zanchetta, G., Wagner, B., Böhm, A., Nowaczyk, N., Rethemeyer, J., and Hilgers, A.: Tephrostratigraphic studies on a sediment core from Lake Prespa in the Balkans, Climate of the Past, 9, 267–287, https://doi.org/10.5194/cp-9-267-2013, 2013.

Hermanns, Y. M., Cortizas, A. M., Arz, H., Stein, R., and Biester, H.: Untangling the influence of in-lake productivity and terrestrial organic matter flux on 4,250 years of mercury accumulation in Lake Hambre, Southern Chile, Journal of Paleolimnology, 49, 563–573, https://doi.org/10.1007/s10933-012-9657-7, 2013.

Horowitz, H. M., Jacob, D. J., Zhang, Y., Dibble, T. S., Slemr, F., Amos, H. M., Schmidt, J. A., Corbitt, E. S., Marais, E. A., and Sunderland, E. M.: A new mechanism for atmospheric mercury redox chemistry: implications for the global mercury budget, Atmos. Chem. Phys., 17, 6353–6371, https://doi.org/10.5194/acp-17-6353-2017, 2017.

Jiskra, M., Wiederhold, J. G., Skyllberg, U., Kronberg, R.-M., Hajdas, I., and Kretzschmar, R.: Mercury Deposition and Re-emission Pathways in Boreal Forest Soils Investigated with Hg Isotope Signatures, Environ. Sci. Technol., 49, 7188–7196, https://doi.org/10.1021/acs.est.5b00742, 2015.

Jiskra, M., Heimbürger-Boavida, L. E., Desgranges, M. M., Petrova, M. V., Dufour, A., Ferreira-Araujo, B., Masbou, J., Chmeleff, J., Thyssen, M., Point, D., and Sonke, J. E.: Mercury stable isotopes constrain atmospheric sources to the ocean, Nature, 597, 678–682, https://doi.org/10.1038/s41586-021-03859-8, 2021.

Jiskra, M., Guédron, S., Tolu, J., Fritz, S. C., Baker, P. A., and Sonke, J. E.: Climatic Controls on a Holocene Mercury Stable Isotope Sediment Record of Lake Titicaca, ACS Earth and Space Chemistry, 6, 346–357, https://doi.org/10.1021/acsearthspacechem.1c00304, 2022.

Jovanovska, E., Cvetkoska, A., Hauffe, T., Levkov, Z., Wagner, B., Sulpizio, R., Francke, A., Albrecht, C., and Wilke, T.: Differential resilience of ancient sister lakes Ohrid and Prespa to environmental disturbances during the Late Pleistocene, Biogeosciences, 13, 1149–1161, https://doi.org/10.5194/bg-13-1149-2016, 2016.

Kurz, A. Y., Blum, J. D., Washburn, S. J., and Baskaran, M.: Changes in the mercury isotopic composition of sediments from a remote alpine lake in Wyoming, USA, Science of the Total Environment, 669, 973–982, https://doi.org/10.1016/j.scitotenv.2019.03.165, 2019.

Leng, M. J., Baneschi, I., Zanchetta, G., Jex, C. N., Wagner, B., and Vogel, H.: Late Quaternary palaeoenvironmental reconstruction from Lakes Ohrid and Prespa (Macedonia/Albania border) using stable isotopes, Biogeosciences, 7, 3109–3122, https://doi.org/10.5194/bg-7-3109-2010, 2010.

Leng, M. J., Wagner, B., Boehm, A., Panagiotopoulos, K., Vane, C. H., Snelling, A., Haidon, C., Woodley, E., Vogel, H., Zanchetta, G., and Baneschi, I.: Understanding past climatic and hydrological variability in the mediterranean from Lake Prespa sediment isotope and geochemical record over the last glacial cycle, Quaternary Science Reviews, 66, 123–136, https://doi.org/10.1016/j.quascirev.2012.07.015, 2013.

Luo, H.-W., Yin, X., Jubb, A. M., Chen, H., Lu, X., Zhang, W., Lin, H., Yu, H.-Q., Liang, L., Sheng, G.-P., and Gu, B.: Photochemical reactions between mercury (Hg) and dissolved organic matter decrease Hg bioavailability and methylation, Environmental Pollution, 220, 1359–1365, https://doi.org/10.1016/j.envpol.2016.10.099, 2017.

Obrist, D., Kirk, J. L., Zhang, L., Sunderland, E. M., Jiskra, M., and Selin, N. E.: A review of global environmental mercury processes in response to human and natural perturbations: Changes of emissions, climate, and land use, Ambio, 47, 116–140, https://doi.org/10.1007/s13280-017-1004-9, 2018.

O'Driscoll, N. J., Vost, E., Mann, E., Klapstein, S., Tordon, R., and Lukeman, M.: Mercury photoreduction and photooxidation in lakes: Effects of filtration and dissolved organic carbon concentration, Journal of Environmental Sciences, 68, 151–159, https://doi.org/10.1016/j.jes.2017.12.010, 2018.

Schaefer, K., Elshorbany, Y., Jafarov, E., Schuster, P. F., Striegl, R. G., Wickland, K. P., and Sunderland, E. M.: Potential impacts of mercury released from thawing permafrost, Nature Communications, 11, 1–6, https://doi.org/10.1038/s41467-020-18398-5, 2020.

Soerensen, Anne. L., Schartup, A. T., Gustafsson, E., Gustafsson, B. G., Undeman, E., and Björn, E.: Eutrophication Increases Phytoplankton Methylmercury Concentrations in a Coastal Sea—A Baltic Sea Case Study, Environ. Sci. Technol., 50, 11787–11796, https://doi.org/10.1021/acs.est.6b02717, 2016.

---

## Author Comment (AC2)

Please find below our detailed responses (in blue) to comments given by Reviewer #2, where the original reviewer comments are repeated here in black for clarity and completeness.

**Reviewer #2**

https://doi.org/10.5194/egusphere-2023-1363-RC2

This study examines long-term Hg records in sedimentary archives due to its sensitivity to centennial to millennial-scale environmental variations. Sediment analysis from two interconnected lakes, Lake Prespa and Lake Ohrid, over the past 90,000 years reveals distinct Hg patterns. Divergent Hg signals during the early and middle Holocene suggest that local factors significantly influence the Hg cycle's response to environmental changes, highlighting the role of sediment composition, lake structure, and water balance in determining the local versus global influences on Hg signals. It is a very interesting topic. This paper contains dense content and thorough analysis with well-written explanations. I am curious about whether the biota species are the same in both lakes, as this could be another factor impacting the differences in Hg records between these two bodies of water. Additionally, the layout of the paper could be improved, such as placing tables and figures at the end of the manuscript, which would enhance its readability and organization.

We give many thanks to Reviewer #2 for their kind and constructive feedback on our manuscript and are thrilled they found it to be an interesting read. In the response below and in our revised manuscript, we will endeavor to ensure the questions raised are addressed, and alterations made where necessary.

**Introduction**

1. Line 35-44 In the first paragraph, I believe it would be good to emphasize the bi-directional pathway of Hg transportation. Hg can not only be emitted/released into the atmosphere but can also deposit into terrestrial and oceanic ecosystems.

   This is a good suggestion. To highlight the bi-directional nature of Hg cycling in the environment, we will add the following text to the manuscript:

   **Lines 35 – 36**: *Mercury (Hg) is a volatile metal released into the atmosphere, lakes, and ocean from both natural and anthropogenic sources, and actively cycled between surface reservoirs.*

   **Lines 50 – 53**: *Evasion back to the atmosphere, consumption by living organisms, or sequestration within aquatic sediment all represent ways in which Hg may 'leave' the terrestrial environment; the latter are known to be particularly effective sinks within the global Hg cycle (Bishop et al., 2020; Selin, 2009).*

   We believe this text would best fit into the second paragraph of the introduction (rather than the first), to ensure clarity for the reader, and ensure reviewer #2's suggestion was suitably integrated into the narrative flow.

2. Line 83, please remove this subtitle as there are no other subtitles in the Introduction section.

   Good point. This subtitle will be removed so the introduction text is presented as one single passage.

3. Line 112, it would be good to include the full name of Hg$_{AR}$, as this is the first instance of its mention in the manuscript.

   We thank reviewer #2 for pointing this out. This sentence will be revised to read:

   **Lines 115 – 118**: *… where (single) host-phase abundance or dilution cannot be easily accounted for, **Hg accumulation rate** (Hg$_{AR}$) may provide the most optimal assessment of Hg availability through time as long as a robust age model is available for the archive.*

4. Fig 2. Study map normally locates in "2. Site Description."

   Following this suggestion, **Figure 2** will be moved to **section 2**.

**Site Description**

5. Line 148 to 149, delete the dashed line.

Good spot, this sentence will be revised to read:

**Lines 148 – 151**: *Major shifts in sedimentation and catchment structure of lakes Prespa and Ohrid generally correspond to the large-scale climate oscillations captured by proxy records across southern Europe throughout the last glacial-interglacial cycle (~100-kyr) (e.g., Rasmussen et al., 2014; Sanchez Goñi and Harrison, 2010; Tzedakis et al., 2006).*

6. Line 259 to 265, I recommend merging and simplifying this content with the information found between lines 207 to 211 and lines 215 and 217.

Another good suggestion. Our initial decision to keep this information separate was in light of the fact that the secondary datasets mentioned here were not obtained as part of the same study, with different teams leading the data acquisition process. Thus, merging the data in our revised manuscript may compromise the clarity of this section. However, we do concur that this information could be presented more concisely, and so will shorten this paragraph by ~10 % through removal of superfluous wording.

7. Line 260 and 261, It appears that the same method is used to calculate TOC, but there are different references compared to lines 210 and 211. Is there a specific reason for this discrepancy?

Yes. Sedimentological analysis of the two cores was carried out at the same institution (University of Cologne) and the method used to calculate TOC was the same. However, TOC values for Lake Prespa and Lake Ohrid were calculated and presented in separate studies:

**Prespa** → (Aufgebauer et al., 2012) (~17–0 ka), and (Damaschke et al., 2013) (~90–0 ka)

**Ohrid** → (Francke et al., 2016)(~600 – 0 ka)

Separation of the two into discrete sections of the manuscript serves to acknowledge this difference, and the corresponding text has been edited to read:

**Lines 216 – 218**: *TOC was calculated as the difference between TC and TIC by Aufgebauer et al. (2012) for the upper ~3.2 m, and by Damaschke et al. (2013) for the full ~17 m succession.*

**Lines 267 – 269**: *The first dataset comprises TC and TIC measured using a DIMATOC 200 (TOC calculated as the difference between TC and TIC), and TS using a Vario Micro Cube combustion CNS elemental analyser at the University of Cologne - both by Francke et al. (2016).*

**Section 3.3 Mercury measurements**

8. Line 292 and 293, why use different resolution to analyze Hg sediment samples from these two lakes?

Sampling and analysis of the two cores were done as separate studies, each with key differences that influenced the sampling strategy. For example, core Co1215 from Lake Prespa was recovered in 2007 (Wagner et al., 2010): to be directly compared to core Co1202 taken from Lake Ohrid (also in 2007) (Vogel et al., 2010), and subsequently facilitate a better understanding of interactions between the two lakes during the last glacial. Given the length of this core and associated chronological interval (<100-kyr), a finer sampling strategy was chosen. Conversely, the 5045-1 core was extracted during an ICDP drilling campaign in spring 2013 (SCOPSCO - Wagner et al., 2014). Although this core is the deepest, most complete paleoenvironmental record from Lake Ohrid currently available, the sampling strategy adopted by the SCOPSCO team was intended to cover the full ~1-Myr succession (16 cm per sample). This study focusses on the upper ~100-kyr of core 5045-1, and thus

explains why the Hg record from this core is lower resolution than Co1215 (Prespa – 2 cm per sample).

9. Line 293 what is the size of the powered samples, homogenize of sediment samples are really important.

A good aspect of detail that should have been included in our original submission, and so we will add the following:

**Lines 299 – 306**: *Samples were analysed for $Hg_T$ at a resolution of ~2 cm for Lake Prespa, and ~16 cm for 5045-1 (see sections* **3.1** *and* **3.2***). Approximately 2 $cm^3$ of sediment was homogenized to fine powder for TOC (previous studies) and Hg analyses (this study). Powdered samples were weighed into glass measuring boats, with masses ranging between 35–96 mg for Co1215, and between 27–78 mg for 5045-1. For samples particularly rich in inorganic fractions (e.g., samples coinciding with tephra layers), masses needed to be greater in order to yield a sufficiently high peak area (Lumex output) for calculation of sediment mercury concentrations: justifying the range in weights for both cores.*

10. Line 296 Could you provide information relating to percent recovery for the standard material?

We were not completely sure what reviewer #2 was alluding to here, and so we provide responses to all possible interpretations of this question, which is a valuable one to ask in both respects.

1) **How much Hg was recovered from the standard material relative to the certified values** → This value is difficult to ascertain based on our analyses alone. Nonetheless, we assume for % recovery to be equal to 100 % (or very close to). Issues with Hg recovery are known to be significantly less problematic in pyrolysis-based analyses (such as we use in this study) compared to those conducted using inductively coupled plasma mass spectrometry (e.g., laser ablation)(Bin et al., 2001), which generally emerge due to several factors:
- Hg has a very high first ionization potential, resulting in low sensitivity as only ions (not atoms) are measured by ICP-MS.
- Hg has seven naturally occurring isotopes; all <30% abundant. Because the total element concentration is divided among many separate isotopes, the number of ions (and therefore the sensitivity) is lower for each individual isotope.
To ensure accuracy of total Hg measurements, we also routinely check the concentration of the standard used in this study with other NIST materials, each with certified Hg contents. For example, NIST 2782 (industrial sludge, 1100 ± 190 ng g$^{-1}$) and NIST 1944 (New Jersey waterway sediment, 3400±500 ng g$^{-1}$).

2) **The average % deviation of standard concentrations from their accepted value** → Across all analytical runs, over 95 % of standards yielded concentrations that were within ±20% of the expected value (here being 290 ng g$^{-1}$), and 68 % were within ±10 %. For both cores, standards with >20 % deviation (11 out of 184 total standard runs) were not used in calibration of the instrument. Details of standard measurements for each record are included as a supplementary Excel file (BGs) SUPPLEMENT_standard runs). In this spreadsheet, we include details of the total number of standards run, standard sample masses, measured Hg concentrations, the peak area derived from varying masses of standard, and the percentage deviation of calculated Hg concentrations from the expected value.

3) **Percentage core material recovered during drilling** → At the DEEP site of Lake Ohrid, six parallel holes yielded 1526m of sediment in total. Accounting for sediment–core overlap, the total composite field recovery amounts to > 95% (545 m). Full details are presented in Wagner et al. (2014), and this information will be added to the manuscript:

**Lines 253 – 255**: *Sediments below 1.5 m depth were recovered from six closely-spaced drill holes at the site in 2013 (5045-1A to 5045-1F), with a total composite field recovery amounting to > 95% (545 m); accounting for sediment–core overlap (Wagner et al., 2014).*

Core recovery from Lake Prespa had not been published at the time of writing, and so we cannot provide a % value. However, we assume this was also high (>95 %) given the lack of any major gaps in sedimentation and/or disturbance of the core samples; likely due (at least

in part) the undisturbed sedimentation at the Co1215 site inferred from hydroacoustic surveillance (Wagner et al., 2010).

11. Line 299 Could you please specify the exact table or figure that indicates the calibration results here?

> As above, details of standard runs for each record are included as a supplementary Excel file ((BGs) SUPPLEMENT_standard runs), and reference to this information will also be incorporated into the revised manuscript as the following statement:
>
> **Line 314***: Details of standard runs for each core are included as a supplementary file.*

12. Line 303 I recommend removing the subtitle 3.3.1 since there are no other subtitles in this section.

> This is a fair suggestion and can understand why it was made here. However, we believe this subtitle serves an important purpose of guiding the reader through this part of the manuscript – as new formulae, data, and methods were introduced as part of our Hg_{AR} calculations. Hence, separating this information into a sub-section helps to guide the reader through our workflow more clearly.

13. Fig 4. The legend for MIS 3-5 in the figure is not easy to identify. It's up to your discretion whether to consider using different colors to improve clarity.

> We agree that the accessibility and clarity of this figure needed improvement; specifically, the presentation of data for MIS 3-5. We include details of our proposed changes, and a copy of the revised **Figure 4** below:
>
> *MIS 5* – changed from circles to plus symbols. Colour changed to lilac.
> *MIS 4* – changed from circles to triangles, with reduced transparency. Colour changed to light blue.
> *MIS 3* – changed from circles to squares. Colour changed to navy blue.
> *MIS 2* – colour changed to black.

[Figure]

**Figure 4**: A comparison of host-phase relationships between lakes Prespa and Ohrid. Points are coded relative to stratigraphic period: the Holocene (12–0 ka, transparent circles), and the late Pleistocene (90–12 ka, filled symbols). We

compare $Hg_T$ records for both lakes relative to total organic carbon (TOC), sulphide (estimated by total sulphur (TS)), and detrital minerals (estimated by potassium (K) concentrations) – note that aluminium (Al) data are more commonly used as an indicator of detrital mineral abundance, but these are currently unavailable for 5045-1.

14. Line 390, what is p-value for the relationship between $Hg_T$ and TS for MIS 1?

The p value for the $Hg_T$/TS relationship in Lake Prespa for MIS 1 is 0.8534. To make p-values for both cores readily accessible to the reader but without adding more quantitative data to the main text, we propose adding a column into supplementary **Table S4**:

**Table S4**: Comparison of host-phase relationships (presented as the r-squared ($r^2$) value) between Lake Prespa and Lake Ohrid. $r^2$ values marked in bold/italic signal that the linear relationships observed between $Hg_T$, and each compound examined was negative.

| Lake | Host | MIS | $r^2$ value | p-value |
|------|------|-----|----------|---------|
| Prespa | $Hg_T$/TOC | 1 | 0.3375 | <0.0001 |
| | | 2 | 0.0105 | 0.7858 |
| | | 3 | 0.1053 | <0.0001 |
| | | 4 | 0.1381 | <0.0001 |
| | | 5 | *0.0002* | *0.8559* |
| | $Hg_T$/TS | 1 | 0.2511 | <0.0001 |
| | | 2 | 0.0007 | 0.8534 |
| | | 3 | 0.056 | 0.0002 |
| | | 4 | 0.0431 | 0.0132 |
| | | 5 | 0.0751 | 0.0001 |
| | $Hg_T$/K | 1 | *0.4418* | *0.7580* |
| | | 2 | *0.0184* | *0.3531* |
| | | 3 | *0.0024* | *0.4390* |
| | | 4 | *0.031* | *0.0362* |
| | | 5 | 0.0109 | 0.1530 |
| Ohrid | $Hg_T$/TOC | 1 | 0.019 | 0.7580 |
| | | 2 | *0.1495* | *0.0006* |
| | | 3 | *0.1477* | *0.0021* |
| | | 4 | *0.0293* | *0.3256* |
| | | 5 | *0.0004* | *0.8976* |
| | $Hg_T$/TS | 1 | *0.007* | *0.1277* |
| | | 2 | *0.0367* | *0.2530* |
| | | 3 | *0.0074* | *0.3750* |
| | | 4 | 0.1197 | 0.0417 |
| | | 5 | 0.0805 | 0.0560 |
| | $Hg_T$/K | 1 | 0.0287 | <0.0001 |
| | | 2 | 0.1574 | 0.0005 |
| | | 3 | 0.1403 | 0.0068 |
| | | 4 | 0.3248 | 0.0004 |
| | | 5 | 0.5239 | <0.0001 |

15. I am wondering if biota species are the same between these two lakes Hg pool/accumulation, you have compared the hydrology, sedimentation regime, and geochemistry of them.

This is a great point. We posit that the link between Hg and biota in lakes Prespa and Ohrid exists as a function of their respective differences in bathymetric structure. In summary:

**Lake Prespa →** shallower (~14 m) waters host a dominantly mesotrophic (nutrient-rich) system where benthic and planktonic diatom species are present in equal abundance and allude to moderate/high biological productivity. We hypothesize that elevated

productivity (inferred from the presence of these species) would typically favor more effective Hg scavenging by organic particles in Lake Prespa, and so could explain why the Hg/TOC relationship is notably stronger in this record compared to lake Ohrid.

**Lake Ohrid** → deep (~240 m) waters of Lake Ohrid host a highly oligotrophic (nutrient poor) environment characterized by low levels of biological productivity, and a high abundance of planktonic diatom species. These conditions would be less favorable for algal scavenging of Hg, and so could explain: (1) why the Hg in Lake Ohrid is inversely correlated to organic matter availability, and (2) why Hg signals better correspond to low-amplitude climate variability rather than transient disturbances.

The role of biotic processes as they relate to Hg could certainly have been described more explicitly within the manuscript; particularly **section 4.4**. Directly inspired by reviewer #2, we will add a paragraph detailing the aforementioned differences to **section 4.4**, with the concluding statement:

> **Lines 719 – 723**: *While the overall signal will remain dominated by Hg availability, broad-scale differences in productivity between lakes Prespa and Ohrid through time could provide an additional explanation for the disparate expression of recorded Hg signals (**section 4.1**); with notably higher productivity in the shallower Lake Prespa further increasing its sensitivity to changes in nutrient status, erosion, and hydrology.*

We will also supplement our interpretation of the Hg profiles with additional references to biological data earlier in the manuscript. For example:

- **Section 4.1,** where we consider the role of changing organic processes in creation and/or preservation of the Hg signals we observe in the two lakes:

  > **Lines 416 – 421**: *On one hand, Hg signals could reflect changes in the dominant sources of organic and detrital materials deposited in the lake. For example, combined isotopic and sedimentological data record episodes of stronger algal blooms during MIS 1 and 5 (Leng et al., 2013), supported by coeval abundance of freshwater diatom genera such as Cyclotella and Aulacoseira (Cvetkoska et al., 2015). All correspond to elevated HgT, and so could imply more effective Hg burial by autochthonous organic material compared to allochthonous (Leng et al., 2013; Damaschke et al., 2013).*

- **Section 4.3**, where we draw upon biotic evidence for lower water levels in Lake Prespa during the LGM to propose why this lake records distinctly different glacial/interglacial signals:

  > **Lines 570 – 576**: *One plausible explanation could be a disproportionately large change in Lake Prespa's total volume compared to Lake Ohrid, with implications for seasonal ice cover. Increased abundance of small Fragilariaceae and benthic Eolimna submuralis diatom species point to generally low temperatures and lake levels during MIS 2 (Cvetkoska et al., 2015), and are reinforced by elevated concentrations of coarse sand and gravel grains (IRD) alluding to persistent ice formation on the lake surface (Damaschke et al., 2013; Wagner et al., 2010).*

A full presentation of the biological character of each lake is beyond the scope of this study, although detailed descriptions of this nature are presented in the following publications - all of which are cited in our manuscript:

> Cvetkoska, A., et al. (2018) Spatial patterns of diatom diversity and community structure in ancient Lake Ohrid. *Hydrobiologia* **819**, 197–215.
>
> Cvetkoska, A., et al. (2016) Ecosystem regimes and responses in a coupled ancient lake system from MIS5b to present: the diatom record of lakes Ohrid and Prespa. *Biogeosciences* **13**, 3147–3162
>
> Cvetkoska, A., et al. (2021) Drivers of phytoplankton community structure change with ecosystem ontogeny during the Quaternary. *Quaternary Science Reviews* **265**, 107046, https://doi.org/10.1016/j.quascirev.2021.107046

Jovanovska, E., et al. (2022) Environmental filtering drives assembly of diatom communities over evolutionary timescales. *Global Ecology and Biogeography* **31**, 954–967

Leng, M. J., et al. (2013) Understanding past climatic and hydrological variability in the Mediterranean from Lake Prespa sediment isotope and geochemical record over the last glacial cycle, *Quaternary Science Reviews* **66**, 123–136

---

## Author Comment (AC3)

Please find below our detailed responses (in blue) to comments given by Reviewer #3, where the original reviewer comments are repeated here in black for clarity and completeness.

**Reviewer #3**

https://doi.org/10.5194/egusphere-2023-1363-RC3

I have completed my review. I have a few suggestions for the manuscript and no basic criticisms. The main conclusion is that there is no similar pattern for the two lakes and there is not a unifying hypothesis to explain the Hg variability. Different attempts to link the Hg flux to different sources fail to be conclusive for all the periods. This can be frustrating, but it is a conclusion.

> We thank reviewer #3 for taking the time to read our work, and for their constructive feedback. We agree that it is somewhat difficult to obtain a conclusive outcome in this study. However, we also feel this outcome reinforces the value of the study, as it provides valuable information on the complexity of Hg cycling through terrestrial lake systems on these timescales. Thus, we hope that it will inspire future work in this research domain.

I think the statistic applied at the data is too basic to be able to give a better understanding (if different handling of the data can produce an easier interpretation), but I'm not so expert to give some further comments on that.

> Exploration of how different statistical methods may be used in analysis of Hg-based datasets would certainly be valuable in the future. We agree the level of statistics here applied is modest - this was done on purpose. While we have tested various other analyses, none of these yielded appreciable improvements or insights. For example, application of robust regression could partially account for non-linearity between Hg and host phase variability in the two datasets, yet it also required selection of one dominant host phase for each core. Distinct anomalies in both records suggest that (i) the dominant host phase did indeed change at discrete points in time and (ii) the total accumulation (availability) of Hg in the system is equally important so that normalized Hg needs to be treated with caution. From this, we concluded that more advanced data treatment is not warranted and could end up obscuring clear assessment of the processes underpinning the observed signals.

I can additionally note that during the Holocene TIC increase and it would be interesting to normalize the data also for TIC for this interval.

> This is certainly an interesting observation. We agree that TIC variability warrants more explicit mention in the manuscript, and so will first make the following text revision:

>> **Lines 616 – 618**: *Between ~12 and 3 (±0.5–0.2 (1σ)) ka Lake Prespa captures a series of large peaks in $Hg_T$ and $Hg_{AR}$, corresponding to high TOC and TIC indicative of elevated productivity, higher rates of organic material preservation, and limited mixing (**Fig. 5**). Conversely, $Hg_T$ and $Hg_{AR}$ show a progressive decline in Lake Ohrid during MIS 1, despite coeval increases in TOC and TIC (**Fig. 6**).*

> Below and in our revised supplementary file (**Figure S5**), we will show a variation on **Figure 5** which includes a profile for Hg normalized to TIC (shaded in yellow). This profile appears remarkably similar to Hg/TOC, with low values shown throughout the Holocene, where clear TIC peaks are visible. This observation is not surprising, given that the TIC in Lake Prespa is generally enriched in organic-rich intervals where algal productivity is high, lake water mixing was reduced, and bottom waters were anoxic for longer periods (Leng et al., 2013). High productivity also explains the presence of endogenic calcite (TIC) during the Holocene, as it would more readily facilitate increased $CO_2$ assimilation; a process enhanced by plentiful bicarbonate supply from surface run-off into the lake, concentrated by enhanced evaporation due to warmer regional temperatures (Leng et al., 2013; Matzinger et al., 2007). Together,

this suggests that the processes underpinning the relatively strong Hg/TOC relationship during the Holocene are also linked to those associated with variability in TIC.

[Figure]

**Figure 5**: Total Hg (Hg$_T$) and total Hg accumulation rate (Hg$_{AR}$) for core Co1215 from Lake Prespa, presented as a function of depth and time, and relative to lithofacies, visible (grey shading) and cryptotephra (orange shading) layers. We include records of Hg$_T$ (this study) normalized to records of total organic carbon (TOC) (Damaschke et al., 2013), sulphide (estimated by total sulphur (TS)) (Aufgebauer et al., 2012), detrital mineral abundance (estimated by potassium (K)) (Panagiotopoulos et al., 2014), and total inorganic carbon (TIC – highlighted in yellow) (Damaschke et al., 2013), with filled shading marking the original datasets. A distinct lake low stand based on seismic profiles and sedimentological data is marked at 14.63 - 14.58 m depth (red shading) (Wagner et al., 2014). A purple arrow marks sections where artificially high Hg/TOC values are generated by a sharp drop to near-zero TOC (<0.06 wt. %) coinciding with deposition of the Y-5 (17.1 m) tephra unit – an effect expected as background sedimentation is interrupted by volcanic ash deposition. White boxes mark the marine isotope stages defined by (Lisiecki and Raymo, 2005), and stratigraphic periods are labelled in black/white.

We suggest that changing **Figure 5** of the main text is not necessary in light of the broad similarity of the Hg/TIC and Hg/TOC profiles during the Holocene and would not constitute a meaningful addition to the manuscript. However, we will make more explicit reference to variability in TIC and it's relation to Hg during the Holocene by means of the revised text presented above, and will include the Figure above in the supplementary file.

Probably, it would be useful to mention in a more direct way tectonic activity as a potential source. The two lakes are placed in a very active tectonic zone, a present large gas emission from fault is present not far from lake Ohrid. Difficult to use this as an argument but it needs to be considered.

This is a welcome suggestion, but as the reviewer also points out, it is difficult to directly address in the absence of quantitative data that can quantify the rate and/or intensity of Hg release from the faults present in the Prespa/Ohrid region. Although these faults were likely active during the most recent ~100-kyr of lake history, earthquake frequency nor intensity has

been reconstructed for this region on these timescales, and so discrete Hg signals cannot be linked to specific events. To acknowledge the potential for Hg emission into the catchment by seismic activity while remaining mindful of manuscript length, we will add an additional section to our attached supplementary file:

**Text SD4: Tectonic activity and Hg release**

Reference to this supplementary discussion will be added to the main text as follows:

> **Lines 704 – 708**: *Local differences in Hg emission by neotectonic activity may have also contributed to the divergent Hg signals, owing to differences in the host rock geology, tectonic instability, and mechanical stress regimes of faults surrounding the two basins (Hoffmann et al., 2010; Lindhorst et al., 2015). However, the significance of these differences cannot be fully assessed in the absence of direct Hg emission measurements (see **Text SD4**).*

An additional argument not considered (which needs at least to be mentioned) is the dust transport. Loess belt is diffuse in the Mediterranean and there is evidence of loess deposition also in Macedonia even if not well described.

To give further credence to evidence for variable dust deposition in the Balkan region over the last glacial period, we will add the following text to the manuscript, with direct reference to the loess-based evidence that reviewer #3 rightly highlights:

> **Lines 538 – 544**: *However, we see no clear evidence atmospheric dust played a major (direct) role in the local Hg cycle in our data. For example, peaks in elemental ratios typically associated with mineral dust deposits (e.g., Zr/Ti) do not correspond to peaks in $Hg_T$ and/or $Hg_{AR}$ (Vogel et al., 2010), nor loess-based evidence substantially higher aeolian dust fluxes over Central Europe and the Balkans during the last glacial maximum (Újvári et al., 2010; Rousseau et al., 2021). Marine sediment records also fail to capture measurable changes in dust fluxes over the Ionian and Aegean seas corresponding to pronounced Hg signals in Lake Ohrid (Ehrmann and Schmiedl, 2021).*

In the paragraph hosting this text, we further describe and assess the potential for a dust-derived Hg source to lakes Prespa and Ohrid during the last glacial. However, given the null result of this exploration, we intend to keep this section relatively short similar to our original submission.

**Page 8**: The description of the climate is not very convincing.

Although we emphasize that the primary aim of this manuscript paper is not a paleoclimate reconstruction, we agree that a more descriptive summary of the regional paleoclimate history would be useful in this section of the manuscript: to provide important context and overview of that may inform the Hg records presented herein. As such, the following changes will be made to the text:

> **Lines 148 – 165**: *Major shifts in sedimentation and catchment structure of lakes Prespa and Ohrid generally correspond to the large-scale climate oscillations captured by proxy records across southern Europe throughout the last glacial-interglacial cycle (~100-kyr) (e.g., Rasmussen et al., 2014; Sanchez Goñi and Harrison, 2010; Tzedakis et al., 2006). Generally higher local temperatures and moisture availability are observed prior to 74 ka, following which conditions became distinctly colder and/or drier. This resulted in the rapid recession of forest ecosystems, intense erosion of local soils and catchments, and elevated aeolian activity. Although slightly warmer conditions were restored between ~57 and 29 ka, both moisture availability and temperature dropped again during the Last Glacial Maximum (LGM; ~29 – 12 ka) – favoring the growth and development of glaciers and (peri)glacial features in the Prespa/Ohrid catchment (Ribolini et al., 2018; Gromig et al., 2018; Ruszkiczay-Rüdiger et al., 2020), but also across the Balkan peninsula (Allard et al., 2021; Hughes and Woodward, 2017; Leontaritis et al., 2020). At ~12 ka, the Pleistocene to Holocene transition saw the rapid propagation of warmer, wetter conditions across the region (known as Termination I) with only brief excursions from this warming trend, such as episodes of transient drying and/or cooling at 8.2 ka and 4.2 ka (Bini et al., 2019; Aufgebauer et al., 2012). Anthropogenic influence on the Balkan landscape becomes increasingly clear from ~2.5 ka onwards, mainly in the form of increased erosion regimes, forest clearance, agricultural land modification, and evidence for metallurgic practices (Panagiotopoulos et al., 2013; Cvetkoska et al., 2014; Radivojević and Roberts, 2021).*

**Line 226**: Be honest is correct to quote Zanchetta et al. 2018 but also Scaillet et al. 2013 QSR 78, 147-154.

We will revise the text to include the correct two citations for this eruption date:

**Line 234**: *Y-6 (45.50 ± 1 ka (Zanchetta et al., 2018; Scaillet et al., 2013))*

**Line 267**: Delete zirconium.

Good observation. Text will be corrected to read:

**Line 272**: Elemental intensities were obtained for K, Ti, Fe, Zr

**Lines 346-348**: This sentence is very vague. By definition Holocene is interglacial. You need to specify e.g., increase of forest and so on.

To make our definition of the Holocene clearer in this section, we will revise the text as follows:

**Lines 364 – 366**: *Widespread proxy-based evidence for warmer temperatures, forest expansion, and increased precipitation representative of interglacial climatic conditions marks the start of the Holocene epoch (~12 ka) in SE Europe…*

**Lines 414-416:** This sentence is obscure to me. Probably useless. What do you mean with "reminiscent". I don't see the importance of this sentence.

We feel that comparison of the Lake Ohrid Hg record with another ancient lake succession does add value to this passage, as it introduces the argument that Hg availability ($Hg_{AR}$) is limited by the Hg fluxes to the catchment and that, even in presence of abundant host-phases, it will not correlate and/or may get diluted instead. However, we agree this sentence could be written more clearly. Thus, we will revise the text to improve the readability of the point being made:

**Lines 444 – 447**: *the relationship between $Hg_T$ and organic matter in Lake Ohrid also shows an inverse correlation (**Fig. 4**), similar to the trend observed in the uppermost sediments of a ~5 Ma succession from Lake Baikal (Russia) (Gelety et al., 2007). These trends may be explained by a scenario where the Hg flux to Ohrid from direct deposition and/or surrounding catchment is typically the limiting factor, rather than availability of potential host phases.*

**Lines 475:** I don't think MIS 3 is considered anymore an interglacial (for a while). I think this sentence should be deleted.

We agree that much of the text here is probably superfluous. While remaining mindful that comparing MIS 3 to the preceding stages serves as a useful frame on which to present our results, but also acknowledging the validity of the point reviewer #3 raises here, the passage will be edited to instead read:

**Lines 507 – 509**: *During MIS 3, proxy records suggest that conditions in the Prespa/Ohrid region were milder than MIS 4, but cooler and drier than MIS 5 (**Fig. 7**) (Panagiotopoulos et al., 2014; Sadori et al., 2016; Wagner et al., 2019)*

**Line 565**: Delete interglacial.

Sentence will be corrected to read:

*Lines 615 – 616: The timing and amplitude of $Hg_T$ and $Hg_{AR}$ signals recorded in Lake Prespa and Lake Ohrid sediments are noticeably different during the Holocene (MIS 1).*

**Lines 560-563**: This seems speculative. Do you have evidence of this in the records?

The reviewer is right in assuming this statement is speculative. It is unclear from our data why the transient Oldest (17.5-14.5 ka) and Younger (12.9-11.7 ka) Dryas climate events did not appear to leave measurable Hg signals in the sediments, and so here we sought to provide a potential explanation. However, something we agreeably did not sufficiently provide in our original submission, are details of how both Lake Prespa and Ohrid record clear indications of these cold events:

**Lake Prespa** → Core Co1215 records a clear, transient change in tree pollen concentrations corresponding to the Younger Dryas. Specifically, studies have revealed an increase in the abundance of cold-resistant open steppe vegetation, which was likely due to a net reduction in winter temperatures and moisture availability in the catchment (Panagiotopoulos et al., 2013; Aufgebauer et al., 2012). A transient shift in diatom flora also support this interpretation, reflecting a nutrient pulse linked to enhanced catchment erosion, lake-level reduction, and wind stress (Cvetkoska et al., 2014).

**Lake Ohrid** → Evidence for Younger Dryas-associated cooling has been identified in core 5045-1 in the form of low tree pollen percentages, low TOC concentrations, and high ($^{234}U/^{238}U$) activity ratios –indicative of cold and dry local conditions, as well as deep hillslope erosion owing in a more open vegetation-type catchment structure (Francke et al., 2019). Short core JO2004-1 extracted from the southern part of the Ohrid basin also records a transient drop in calcite precipitation corresponding to the Younger Dryas, alluding to distinctly colder local temperatures (Lézine et al., 2010).

Inspired by reviewer #3's question, we will better integrate this evidence into the manuscript text as follows:

*Lines 602 – 612: Both lakes contain clear evidence for an abrupt return to glacial conditions during this time. Lake Prespa sediments record shifts in tree pollen and diatom assemblages alluding to a net reduction in local winter temperatures and moisture availability (Aufgebauer et al., 2012; Panagiotopoulos et al., 2013; Cvetkoska et al., 2014), and high uranium ($^{234}U/^{238}U$) activity ratios, low tree pollen percentages, and low TIC concentrations in Lake Ohrid also pertain to intense hillslope erosion owing to a more open catchment structure (Francke et al., 2019; Lézine et al., 2010). Geomorphological evidence also pertains to local glacier stabilization (Gromig et al., 2018; Ribolini et al., 2018; Ruszkiczay-Rüdiger et al., 2020) (**Fig. 7**). Nonetheless, we suggest these events may have been too (a) short-lived, and/or (b) climatically mild to produce a similarly distinct response in the terrestrial Hg cycle as the processes operating during, and immediately following, the LGM; potentially explaining the lack of an associated sedimentary Hg signal.*